# CD1d-dependent immune suppression mediated by regulatory B cells through modulations of iNKT cells

K. Oleinika[1,2], E.C. Rosser[1,3], D.E. Matei[1], K. Nistala[1], A. Bosma[1], I. Drozdov[4] & C. Mauri [1]

Regulatory B cells (Breg) express high levels of CD1d that presents lipid antigens to invariant natural killer T (iNKT) cells. The function of CD1d in Breg biology and iNKT cell activity during inflammation remains unclear. Here we show, using chimeric mice, cell depletion and adoptive cell transfer, that CD1d–lipid presentation by Bregs induces iNKT cells to secrete interferon (IFN)-γ to contribute, partially, to the downregulation of T helper (Th)1 and Th17-adaptive immune responses and ameliorate experimental arthritis. Mice lacking CD1d-expressing B cells develop exacerbated disease compared to wild-type mice, and fail to respond to treatment with the prototypical iNKT cell agonist α-galactosylceramide. The absence of lipid presentation by B cells alters iNKT cell activation with disruption of meta-bolism regulation and cytokine responses. Thus, we identify a mechanism by which Bregs restrain excessive inflammation via lipid presentation.

[1] Centre for Rheumatology, Division of Medicine, University College London, London WC1E 6JF, UK. [2] Division of Infection and Immunity, University College London, London WC1E 6BT UK, UK. [3] Infection, Inflammation and Rheumatology Section, Infection, Immunity and Inflammation Programme, UCL Great Ormond Street Institute of Child Health, University College London, London WC1N 1EH, UK. [4] Bering Limited, London TW2 6EA, UK. Correspondence and requests for materials should be addressed to C.M. (email: c.mauri@ucl.ac.uk)

Regulatory B cells (Breg) are effectors of immune tolerance[1]. Although the hallmark of Breg function is the production of interleukin (IL)-10[2], additional Breg-mediated suppression mechanisms include transforming growth factor-β (TGF-β)[3], IL-35[4] release, and PD-L1 expression[5]. Bregs express different surface markers, including CD21, CD23, CD24, CD5, T cell immunoglobulin and mucin domain (TIM)-1, and CD138[6]. A marker that is expressed by the majority of reported Breg subsets, in both mice and humans, is CD1d[1,7]. Yet, the functional relevance of CD1d for Breg-suppressive function remains to be elucidated.

CD1d is a major-histocompatibility-complex (MHC) class-I-like molecule, which presents self-lipid and non-self-lipid antigens to invariant natural killer T (iNKT) cells[8]. Following engagement of the invariant T cell receptor (iTCR) by CD1d–lipid complexes, iNKT cells proliferate, produce cytokines, and become cytotoxic, regulating innate and adaptive immune responses[9]. iNKT cells are involved in the enhancement of anti-tumor immunity, protection against infections, and regulation of autoimmunity[10]. In the latter context, administration of α-galactosylceramide (α-GalCer), the prototypical iNKT cell glycolipid agonist, has been shown to suppress the development of autoimmunity in mice[11–13]. In humans, numerical and functional defects in iNKT cells have been reported in systemic lupus erythematosus (SLE)[1,14,15], rheumatoid arthritis (RA)[14–16], and multiple sclerosis[17]. If and how decreased iNKT cell number or function contributes to autoimmunity remains unknown.

While α-GalCer presentation by B cells to iNKT cells results in the differentiation of antibody-producing B cells by a feedback mechanism[18,19], whether Bregs by interacting with iNKT cells condition their responses remains less explored. We have shown that B cells from SLE patients with active disease express decreased levels of CD1d and do not support the expansion and activation of iNKT cells upon in vitro stimulation with α-GalCer[1]. In SLE patients responding to B cell-depletion therapy, where a repopulation in naive and transitional B cells with regulatory function is reported[20,21], the CD1d recycling defect on B cells was reversed. iNKT cell frequency and function are normalized in the peripheral blood of these patients, suggesting a B-iNKT cell interaction[1]. These results raise two questions: can Bregs instruct iNKT cells with suppressive function, and does the impaired CD1d+ Breg lipid presentation to iNKT cells exacerbate autoimmune responses?

Here, we report a role for CD1d+T2-MZP Bregs in the differentiation of suppressive iNKT cells that restrain excessive arthritogenic T helper (Th)1/Th17 responses, partially via the production of interferon (IFN)-γ. The induction of promyelocytic leukemia zinc finger (PLZF)+IFN-γ+ iNKT cells in response to α-GalCer treatment occurs only in CD1d B cell-competent mice and is independent of B cell production of IL-10. CD1d–lipid presentation by B cells is required to fine-tune the expression of genes involved in cytokine responses, metabolism, and other pathways related to immune cell activation and effector function in iNKT cells. We also demonstrate that, upon MZ B cell depletion, T2-MZP B cells continue to induce the differentiation of immunosuppressive iNKT cells. Our study suggests that lipid presentation by Bregs, and consequent induction of suppressive iNKT cells, could be a way to increase immune tolerance for the control of autoimmune diseases.

## Results

**B cells mediate α-GalCer amelioration of arthritis**. To address the role that B cells play in α-GalCer-iNKT cell-driven amelioration of arthritis, B cell-deficient (µMT) and WT mice were immunized to induce arthritis and treated with α-GalCer (to promote interactions with CD1d-expressing cells). α-GalCer treatment suppressed the development of arthritis in WT mice, while it failed to inhibit the disease in µMT mice (Fig. 1a). Clinical scores were recorded up to either day 3 or day 7 according to the experimental design. An example of disease development in WT mice is shown in Supplementary Fig. 1a. In response to α-GalCer, we observed an equal increase in the frequencies and absolute numbers of iNKT cells in the spleen (Fig. 1b), inguinal lymph nodes, and the liver of µMT and WT mice (Supplementary Fig. 1b–c). Splenic iNKT cells from both α-GalCer-treated µMT and WT mice expressed comparable levels of Ki-67, suggesting that α-GalCer-driven proliferation of iNKT cells was normal in the absence of B cells (Fig. 1c).

PLZF is responsible for the innate-like effector phenotype of iNKT cells and rapid cytokine production, including IFN-γ[22,23], which has also been associated with iNKT cell suppressive function[24–27]. We observed that in the absence of B cells, iNKT cell ability to upregulate the expression of PLZF and IFN-γ in response to α-GalCer was significantly impaired compared to WT mice in the spleen ($p < 0.01$ and $p < 0.001$ by one-way analysis of variance (ANOVA)) (Fig. 1d, e). This is in contrast to the liver where α-GalCer-induced iNKT cells expressed comparable levels of PLZF and IFN-γ in µMT and WT mice (Fig. 1f, g). We also report a reduction of IL-4-expressing iNKT cells in response to α-GalCer in µMT mice compared to WT mice. The frequency of IL-4+ iNKT cells was overall lower than that of the IFN-γ+ iNKT cells. IL-10, IL-13, and GM-CSF were not differentially expressed under the same experimental conditions (Supplementary Fig. 1d). We could not detect FoxP3+[28] or E4BP4+ iNKT cells[29], previously ascribed with regulatory function in the lymph nodes and adipose tissue, respectively, in the spleens of α-GalCer-treated mice (Supplementary Fig. 1e, f).

µMT mice lack B cells from birth, which could impair the development of iNKT cells. To explore the role of B cells in lipid presentation in an environment where iNKT cells have developed in the presence of B cells, adult WT mice were treated either with a depleting anti-CD20 or isotype-control antibody. Mice were then immunized and treated with α-GalCer. The administration of anti-CD20 antibody depleted over 98.9% of B cells in the spleens of mice (Fig. 2a). In agreement with the results observed in µMT mice, α-GalCer treatment ameliorated the disease in control but not in B cell-depleted mice (Fig. 2b). The frequencies and numbers of iNKT cells were similar between B cell-depleted and control mice, with or without α-GalCer treatment (Fig. 2c). A significant increase in the expression of PLZF and IFN-γ in iNKT cells, following treatment with α-GalCer, was observed in control mice ($p < 0.01$ and $p < 0.0001$ by one-way ANOVA), but their upregulation was significantly lower in B cell-depleted mice ($p < 0.01$ and $p < 0.0001$ by one-way ANOVA) (Fig. 2d, e).

As CD11c+ dendritic cells (DC) play an important role in lipid presentation and iNKT cell priming, next, we selectively depleted DCs and assessed their effect on iNKT cells in AIA. Diphtheria toxin was administered to mice that express the diphtheria toxin receptor (DTR) under the control of the *Cd11c* promoter[31]. Due to the important role that DCs play in the early phase of arthritis induction, α-GalCer was, in this instance, administered 8 h after intra-articular injection of mBSA and intraperitoneal administration of diphtheria toxin. α-GalCer-mediated suppression of arthritis in CD11c+ cell-depleted mice was equivalent to control mice (Supplementary Fig. 2a–c). The upregulation of PLZF and the early "burst" of IFN-γ by iNKT cells in response to α-GalCer was not affected by the lack of DCs (Supplementary Fig. 3a, b). While there was a numerical reduction of iNKT cells in CD11c+ cell-depleted mice compared to controls at 72 h, this was observed following the peak IFN-γ production at 16 h, when the iNKT cell numbers were comparable (Supplementary Fig. 3c, d).

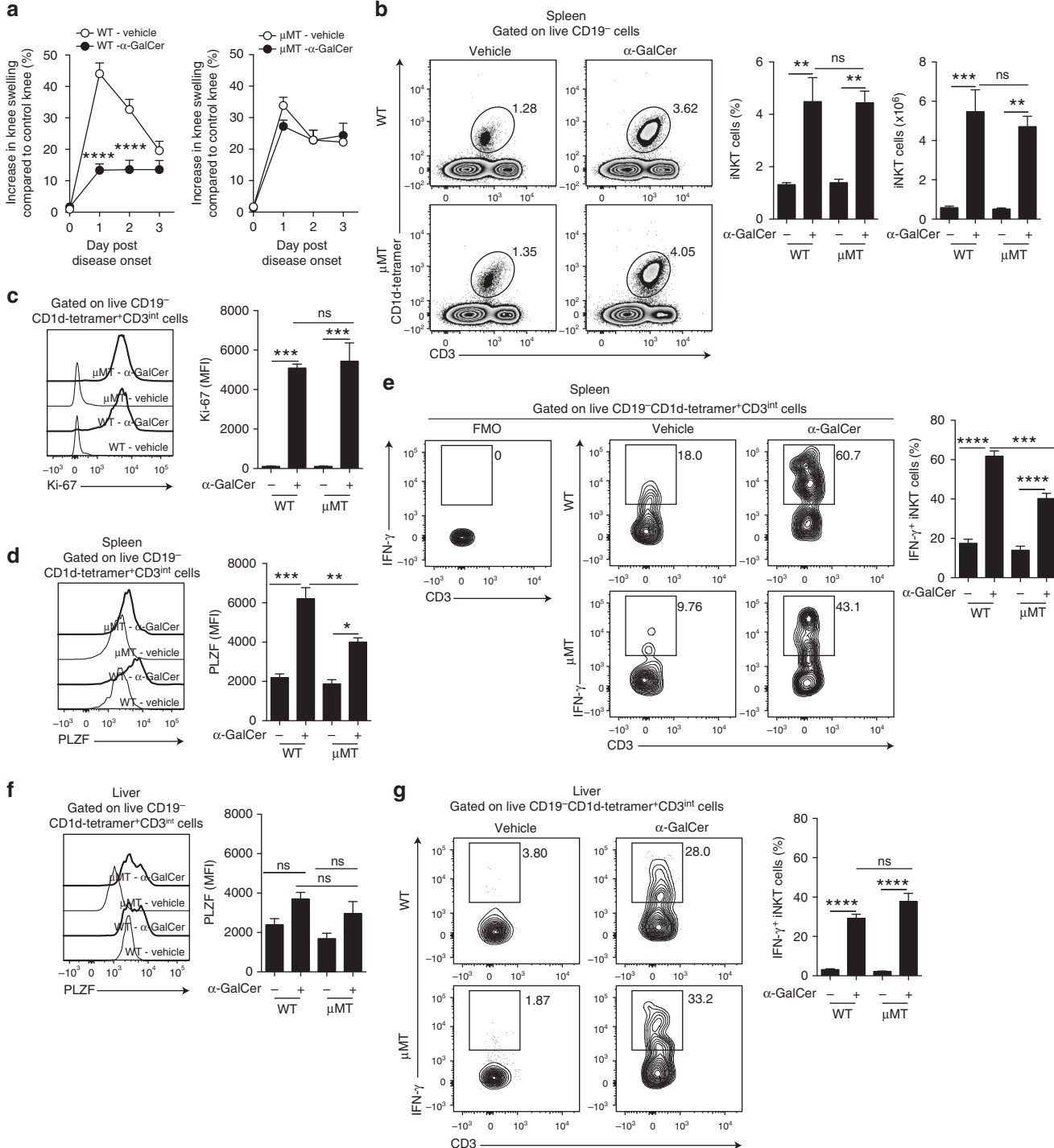

**Fig. 1** α-GalCer-iNKT-cell-dependent amelioration of arthritis requires B cells. **a** Mean clinical score of wild-type (WT) mice (left) and μMT mice (right) that received α-GalCer or vehicle alone following induction of arthritis. The Y axis shows the percentage of swelling in antigen-injected knee compared to control knee (μMT ± α-GalCer $n = 14$ per group, WT ± α-GalCer $n = 20$ per group, one of three experiments is shown). **b** Representative flow cytometry plots and bar charts showing the frequency and number of splenic iNKT cells in μMT and WT mice that received α-GalCer or vehicle alone ($n = 5$ per group, one of three experiments is shown). **c** Representative histograms and bar chart showing median fluorescence intensity (MFI) of Ki-67 in splenic iNKT cells from α-GalCer- or vehicle-treated μMT and WT mice ($n = 3$ per group, one of three experiments is shown). **d** Representative histograms and bar chart showing MFI of PLZF in splenic iNKT cells from α-GalCer- or vehicle-treated μMT and WT mice ($n = 3$ per group, one of three experiments is shown). **e** Representative flow cytometry plots and bar chart showing the frequency of IFN-γ$^+$ splenic iNKT cells from α-GalCer- or vehicle-treated μMT and WT mice ($n = 4$ per group, one of three experiments is shown). **f** Representative histograms and bar chart showing MFI of PLZF in hepatic iNKT cells from α-GalCer- or vehicle-treated μMT and WT mice ($n = 3$ per group, one of two experiments is shown). **g** Representative flow cytometry plots and bar chart showing the frequency of IFN-γ$^+$ hepatic iNKT cells from α-GalCer- or vehicle-treated μMT and WT mice ($n = 4$ per group, one of three experiments is shown). **b**, **c** Analyzed on day 3 post disease onset, **d–g** analyzed at 16 h post disease onset. Data are mean ± s.e.m. ns not significant, $*p < 0.05$, $**p < 0.01$, $***p < 0.001$, and $****p < 0.0001$ (**a** two-way analysis of variance (ANOVA), **b–g** one-way ANOVA)

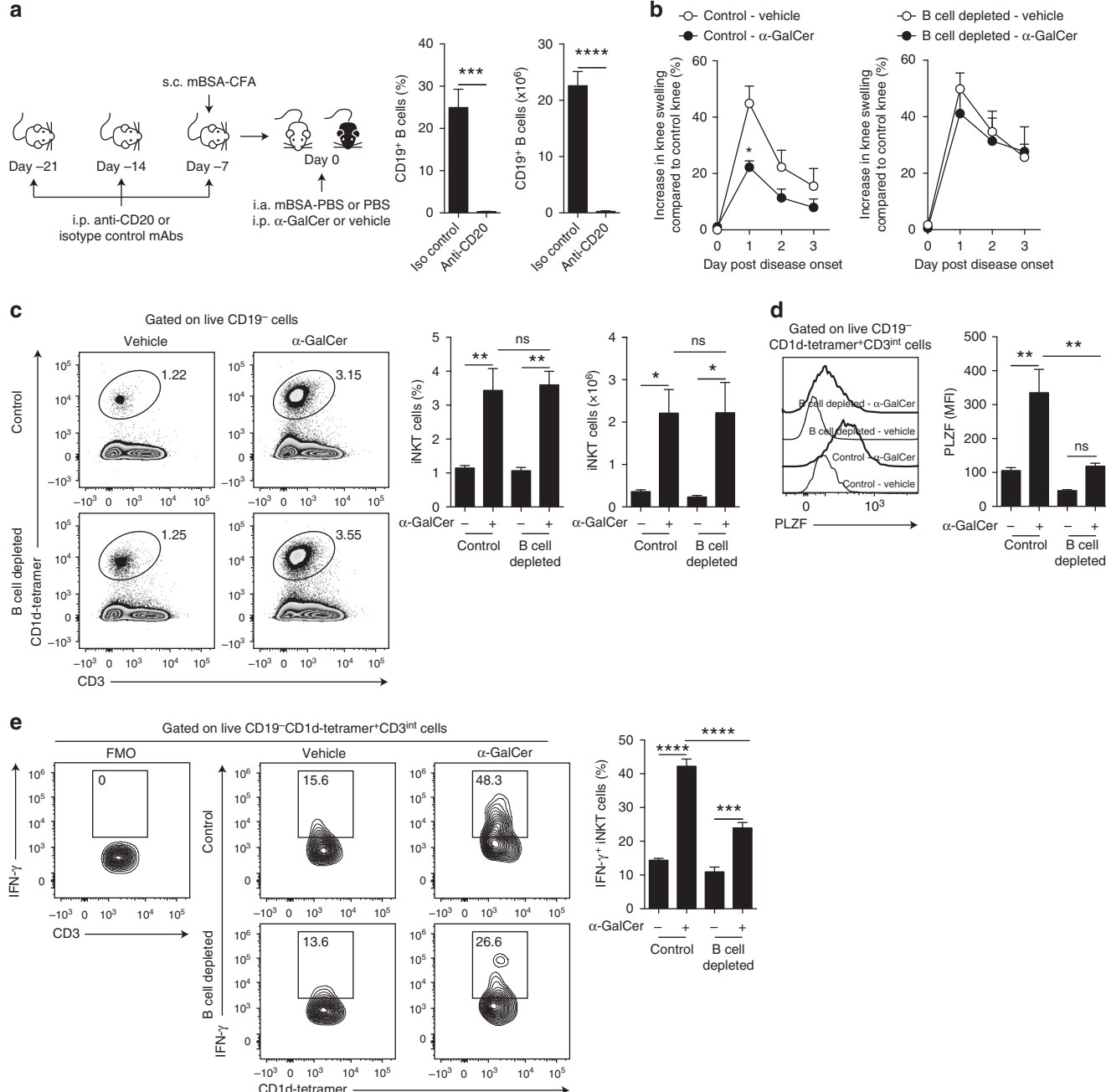

**Fig. 2** B cell depletion abrogates the protective effect of α-GalCer on arthritis. **a** Left, schematic showing the experimental design for B cell depletion. Right, bar charts showing the frequency and number of splenic CD19[+] B cells in B cell-depleted and isotype-control-treated WT mice prior to α-GalCer or vehicle administration ($n = 5$ per group, one of three experiments is shown). **b** Mean clinical score of isotype-control-treated WT mice (left) and B cell-depleted WT mice (right) that received α-GalCer or vehicle alone following induction of arthritis. The $Y$ axis shows the percentage of swelling in antigen-injected knee compared to control knee ($n = 5$ per group, one of two experiments is shown). **c** Representative flow cytometry plots and bar charts showing the frequency and number of splenic iNKT cells in B cell-depleted and isotype-control-treated WT mice that received α-GalCer or vehicle alone ($n = 5$ per group, one of two experiments is shown). **d** Representative histograms and bar chart showing MFI of PLZF in splenic iNKT cells in B cell-depleted and isotype-control-treated WT mice that received α-GalCer or vehicle alone ($n = 3$ per group, one of two experiments is shown). **e** Representative flow cytometry plots and bar chart showing the frequency of IFN-γ[+] splenic iNKT cells in B cell-depleted and isotype-control-treated WT mice that received α-GalCer or vehicle alone ($n = 4$ per group, one of two experiments is shown). **c** Analyzed on day 3 post disease onset; **d**, **e** analyzed at 16 h post disease onset. Data are mean ± s.e.m. ns not significant, $*p < 0.05$, $**p < 0.01$, $***p < 0.001$, and $****p < 0.0001$ (**a** Student's $t$ test, **b** two-way ANOVA, and **c**–**e** one-way ANOVA)

Of interest, the induction of iNKT cell IFN-γ production precedes their expansion (Supplementary Fig. 3e, f).

Next, we depleted B cells or DCs from splenocytes of arthritic mice before stimulating them in vitro with α-GalCer and IL-2. Only B cell depletion led to reduced iNKT cell PLZF and IFN-γ

expression, compared to control and DC-depleted splenocytes. DC depletion led to decreased expression of CD25 and CD69 by iNKT cells (Supplementary Fig. 3g). These data show that while B cells are responsible for driving the expression of iNKT cell PLZF

and IFN-γ, CD11c[+] DCs are required for iNKT cell expansion and activation.

**CD1d[+] B cells induce IFN-γ[+]PLZF[+] iNKT cells**. To ascertain whether B cells interact with iNKT cells directly via CD1d, we generated chimeric mice lacking CD1d on B cells (B-*Cd1d*[−/−]) and control chimeras (B-WT) (Supplementary Fig. 4a). B-*Cd1d*[−/−] mice developed exacerbated disease compared to B-WT mice, suggesting a role for endogenous lipid presentation by B cells in controlling the severity of arthritis (Supplementary Fig. 4b). To assess the importance of B cells expressing CD1d in the differentiation of iNKT cells that limit the disease, we treated B-*Cd1d*[−/−] and B-WT mice with α-GalCer. While α-GalCer suppressed disease development in B-WT mice compared to vehicle-treated B-WT mice, B-*Cd1d*[−/−] mice failed to respond to the treatment (Fig. 3a). α-GalCer-induced inhibition of the disease in B-WT mice was accompanied by a reduction in the frequency and absolute number of IFN-γ[+] and IL-17[+]CD4[+] T cells (Fig. 3b). The frequency of IFN-γ[+] and IL-17[+]CD4[+] T cells on days 3 and 7 is comparable (Supplementary Fig. 4c).

iNKT cell expansion and proliferation in response to α-GalCer were comparable in B-*Cd1d*[−/−] and B-WT mice (Fig. 3c, d). Upon α-GalCer treatment, we observed a significant decrease in PLZF expression by iNKT cells and a decrease in the frequency of IFN-γ[+]PLZF[+] iNKT cells in B-*Cd1d*[−/−] compared to B-WT mice ($p < 0.01$ and $p < 0.05$ by one-way ANOVA) (Fig. 3e, f). We report a decrease in the MFI value of T-bet and an increase in GATA3 expression in iNKT cells from B-*Cd1d*[−/−] compared to B-WT mice in response to α-GalCer (Fig. 3g). These data show that B cell CD1d is critical for the induction of iNKT cells that restrain arthritogenic responses.

It has been previously suggested that iNKT cells regulate the germinal center (GC) entry of B cells and that this is CD1d[+] B cell-dependent[32]. Amelioration of the disease by α-GalCer in B-WT mice was accompanied by a decrease in the number of GCs compared to vehicle-treated mice (Supplementary Fig. 4d). This reduction was controlled by CD1d[+] B cells, since α-GalCer-driven decrease in GCs was not observed in B-*Cd1d*[−/−] mice. In WT mice, α-GalCer treatment caused a reduction in Tfh cells compared to the vehicle-treated group, whereas iNKTfh cells were not significantly changed between the two groups ($p > 0.05$ by Student's *t* test) (Supplementary Fig. 4e). Following α-GalCer treatment, iNKT cells were found in close proximity to IgD[hi]B cells in the T cell area of the spleen in B-WT mice, but not in B-*Cd1d*[−/−] mice (Fig. 3h).

To assess whether in addition to CD1d, Bregs required IL-10 to induce suppressive iNKT cells, we generated mice lacking IL-10-producing B cells (B-*Il10*[−/−]) and WT (B-WT) chimeras (Supplementary Fig. 4a). α-GalCer ameliorated the disease in B-*Il10*[−/−] and B-WT mice (Fig. 3i). No difference between the expression of CD1d on splenic B cells or on other cells between B-*Il10*[−/−] and B-WT mice was observed (Supplementary Fig. 5a). We found that α-GalCer treatment ameliorated arthritis in global *Il10ra*[−/−] mice, just as in WT mice (Supplementary Fig. 5b). These data suggest that IL-10 is not required for the differentiation of suppressive iNKT cells.

**CD1d[+] B cells control iNKT cell transcriptional profile**. To understand the mechanism underlying the suppressive capacity of iNKT cells conferred by B cell lipid presentation, we profiled the transcriptomes of iNKT cells sorted from α-GalCer- or vehicle-treated B-*Cd1d*[−/−] and B-WT mice using RNA-seq. iNKT cell purity was >95% (Supplementary Fig. 6). α-GalCer had similar effects on the majority of gene expression in iNKT cells from B-*Cd1d*[−/−] and B-WT mice (Fig. 4a). However, 3826 genes were

significantly differentially expressed in α-GalCer-treated B-*Cd1d*[−/−] and B-WT mice ($p < 0.05$ by QL F test). Among these were those encoding immune-related genes essential for T and iNKT cell effector function (Supplementary Fig. 7a). Of interest, *Zbtb16*, which encodes PLZF, was expressed at significantly higher levels in iNKT cells from B-WT than from B-*Cd1d*[−/−] mice following α-GalCer treatment ($p < 0.05$ by QL F test), corroborating the results shown in Fig. 3e.

To understand the aspects of iNKT cell function regulated by B cell lipid antigen presentation, we enriched differentially expressed genes from α-GalCer-treated B-*Cd1d*[−/−] and B-WT mice for overrepresented pathways. This analysis revealed gene sets encoding products involved in the cell cycle, NK cell-mediated cytotoxicity, as well as cytokine–cytokine receptor interaction (Fig. 4b).

A substantial increase in the bioenergetic demands over the resting state is a defining feature of T cell activation[33,34]. Analysis of metabolic pathways showed that in α-GalCer-treated B-*Cd1d*[−/−] mice, iNKT cells were highly anabolic and presented an increase in genes encoding enzymes involved in glycolysis and a reduction in those involved in fatty acid oxidation compared to B-WT mice with α-GalCer treatment (Fig. 4c). In particular, crucial mediators of the metabolic switch in effector T cells, namely *Irf4*, *Myc*, and *Hif1a*[33,34], were significantly increased in B-*Cd1d*[−/−] compared to B-WT mice following α-GalCer treatment ($p < 0.05$ by QL F test) (Supplementary Fig. 7a).

iNKT cells have been assigned, similarly to T helper cells, into subsets according to their transcription factor and cytokine profile[35,36]. To further elucidate whether lipid presentation by B cells skewed the response of iNKT cells toward a specific effector subset, we compared the expression of cytokine genes by iNKT cells from B-*Cd1d*[−/−] and B-WT mice with or without α-GalCer treatment. The expression of most cytokines was enhanced by the absence of CD1d-expressing B cells in α-GalCer-treated mice, suggesting that B cells have a different role in iNKT cell activation compared to other antigen-presenting cells. For instance, we found an increase of *Il21*, *Il9*, *Il10*, and *Il17a* in iNKT cells from B-*Cd1d*[−/−] compared to B-WT mice following α-GalCer treatment (Fig. 4d). Increased frequencies of IL-10[+] and IL-13[+] iNKT cells in the absence of CD1d-expressing B cells in α-GalCer-treated mice were also detected by intracellular staining, whereas no positive staining in iNKT cells was found for IL-9, IL-17, and IL-21 (Supplementary Fig. 7b, c).

The transcriptional profiles of iNKT cells isolated from B-*Cd1d*[−/−] and B-WT mice treated with α-GalCer or vehicle control did not overlap with those previously assigned to sorted thymic NKT1, NKT2, and NKT17 cells[36] (Supplementary Fig. 7d). However, at transcriptome-wide level, changes have to be extensive to establish similarity. In addition, in our experiments, iNKT cells were obtained from the spleens of mice during acute inflammation, rather than from the thymus at steady state as previously reported. Therefore, we compared the transcripts that were identified as most significantly enriched in NKT1, NKT2, and NKT17 cells relative to the other subsets. Among significant genes that were differentially expressed ($p < 0.05$ by QL F test), there was an overall skew away from NKT1 and toward NKT2 phenotype in B-*Cd1d*[−/−] mice compared to B-WT mice after α-GalCer treatment (Fig. 4e). This change in transcriptional profile was consistent with the results reported in Fig. 3g showing a lack of upregulation of T-bet, and an upregulation of GATA3 expression in B-*Cd1d*[−/−] mice compared to B-WT mice following α-GalCer treatment. A low number of genes, previously associated with NKT17 cells, were significantly expressed in iNKT cells from B-*Cd1d*[−/−] mice compared to B-WT mice following α-GalCer treatment ($p < 0.05$ by QL F test) (Supplementary Fig. 7e).

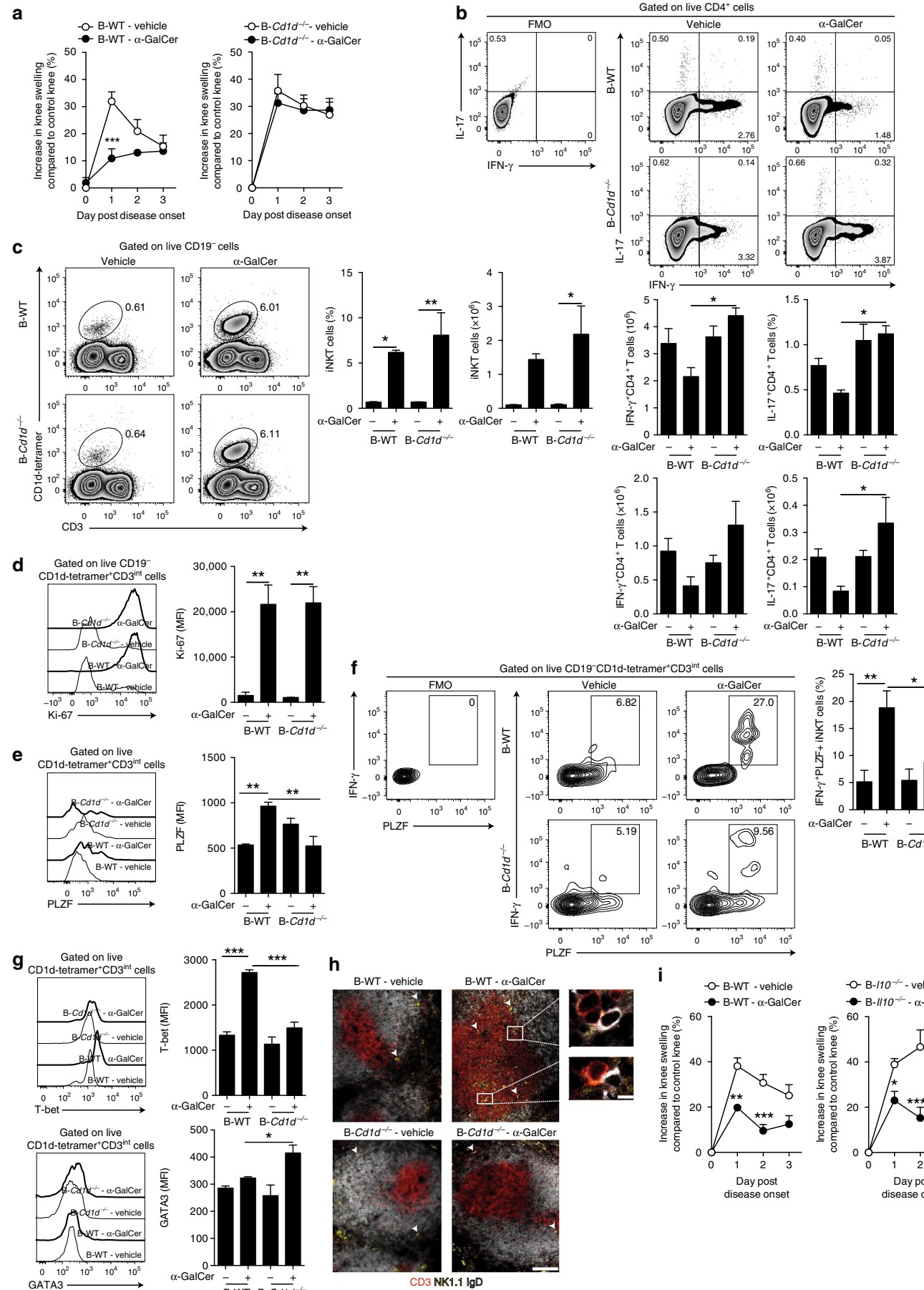

We have consistently shown that upon α-GalCer treatment, the expression of PLZF by iNKT cells from B cell-deficient or B-$Cd1d^{-/-}$ mice is reduced compared to the control group. Therefore, we compared our α-GalCer-treated data sets to 99 genes previously identified as PLZF targets and found that 39 of these transcripts were significantly differentially expressed ($p < 0.05$ by QL F test), with 36 downregulated in the absence of B cells expressing CD1d. This is consistent with an induction of an iNKT cell PLZF signature supported by B cell CD1d. Of interest, $Bach2$, a known repressor of effector differentiation in naive T cells[37], and previously shown to be bound to and repressed by PLZF, was significantly upregulated in iNKT cells of B-$Cd1d^{-/-}$ mice compared to B-WT mice after α-GalCer administration ($p < 0.05$ by QL F test) (Fig. 4f). Thus, iNKT cell suppression of arthritis is dependent on B cell-driven calibration of iNKT cell cytokine response, as well as pathways in iNKT cells associated with effector function and metabolism.

**iNKT cells inhibit Th1 and Th17 responses partially via IFN-γ.** Our results show the inhibition of Th1 and Th17 responses associated with increased IFN-γ production by iNKT cells, and it has been demonstrated that IFN-γ can be protective in auto-immunity, including in AIA[38]. Thus, we investigated whether IFN-γ produced by iNKT cells inhibits Th1 and Th17 responses in our model. We neutralized IFN-γ in vivo[24] prior to α-GalCer or vehicle treatment, aiming to block α-GalCer-driven iNKT cell IFN-γ. α-GalCer-treated control mice, which were protected from the disease (Fig. 5a), displayed lower adaptive IFN-γ and IL-17 production compared to control mice that received vehicle (Fig. 5b and Supplementary Fig. 8). In contrast, neutralization of IFN-γ reduced the suppressive effect of α-GalCer on clinical disease and prevented α-GalCer treatment from reducing the frequency and number of Th1 and Th17 cells. This supports the notion that α-GalCer-induced innate production of IFN-γ has a role in the protection from arthritis and inhibition of adaptive responses.

To understand whether B cell-driven IFN-γ by iNKT cells plays a role in suppressing Th1 and Th17 cells, we used an in vitro suppression assay in which splenocytes, isolated from arthritic μMT or WT mice, were challenged with anti-CD3 (to stimulate CD4$^+$ T cells) and α-GalCer in the presence of anti-IFN-γ. We observed significant downregulation in the expression of T-bet and ROR-γt (transcription factors for Th1 and Th17 cells,

respectively) in CD4$^+$ T cells from WT mice ($p < 0.05$ by one-way ANOVA), but not in CD4$^+$ T cells from μMT mice after α-GalCer stimulation (Fig. 5c). Neutralization of IFN-γ abolished the α-GalCer-driven suppression of T-bet and ROR-γt in WT CD4$^+$ T cells. No significant effect on GATA3 expression in CD4$^+$ T cells from WT or μMT mice by α-GalCer was observed ($p > 0.05$ by one-way ANOVA) (Fig. 5c).

To ascertain whether iNKT cells directly inhibit pathogenic T cells, iNKT cells were isolated from arthritic α-GalCer-treated μMT or WT mice, and then cultured with CD4$^+$ T cells, isolated from arthritic WT mice, in the presence of anti-IFN-γ or isotype-control antibodies. WT and not μMT iNKT cells significantly inhibited WT CD4$^+$ T cell expression of IFN-γ ($p < 0.05$ by one-way ANOVA) (Fig. 5d). The inhibitory effect of WT iNKT cells on CD4$^+$ T cells was abolished upon IFN-γ neutralization in vitro.

Given that blocking IFN-γ, in vivo and in vitro, ablated the ability of α-GalCer-stimulated iNKT cells to suppress Th1 and Th17 cells, we next assessed whether IFN-γ could directly inhibit the responses of Th1 and Th17 cells in vitro. We stimulated CD4$^+$ T cells, isolated from spleens of mice with arthritis, with anti-CD3 and increasing doses of IFN-γ. The frequency of both T-bet$^+$ and ROR-γt$^+$CD4$^+$ T cells was significantly reduced as the concentrations of IFN-γ were increased ($p < 0.001$ and $p < 0.0001$ by one-way ANOVA) (Fig. 5e). These results support the notion that Breg-induced innate production of IFN-γ by iNKT cells has a role in the suppression of inflammation and inhibition of adaptive Th1/Th17 responses.

**T2-MZP Bregs via CD1d induce suppressive iNKT cells.** Based on our findings showing that B cells are important in mediating suppression conferred by α-GalCer, we hypothesized that T2-MZP B cells, known to express high levels of CD1d (Supplementary Fig. 9a–c) and to suppress different autoimmune diseases[39], induced the differentiation of iNKT cells that ameliorate the disease. We adoptively transferred T2-MZP B cells (the purity of sorted subsets is shown in Supplementary Fig. 10), isolated from the spleens of WT mice in remission from arthritis, into WT and $Cd1d^{-/-}$ mice that lack iNKT and type II NKT cells[40,41]. As previously shown, T2-MZP Bregs suppressed arthritis and reduced the frequency and absolute number of IFN-γ$^+$ and IL-17$^+$CD4$^+$ T cells in WT mice; however, WT T2-MZP Bregs were unable to convey the same suppressive effect in iNKT cell-

**Fig. 3** B cell-specific CD1d is critical for α-GalCer-iNKT cell amelioration of arthritis. **a** Mean clinical score of B-WT chimeric mice (left) and B-$Cd1d^{-/-}$ chimeric mice (right) that received α-GalCer or vehicle alone following induction of arthritis. The Y axis shows the percentage of swelling in antigen-injected knee compared to control knee (B-$Cd1d^{-/-}$ ± α-GalCer $n = 5$ per group, B-WT ± α-GalCer $n = 4$ per group, one of three experiments is shown). **b** Representative flow cytometry plots and bar charts showing the frequency (top) and number (bottom) of splenic IFN-γ$^+$ and IL-17$^+$ CD4$^+$ T cells in B-$Cd1d^{-/-}$ and B-WT chimeric mice that received α-GalCer or vehicle alone ($n = 3$ per group, one of two experiments is shown). **c** Representative flow cytometry plots and bar charts showing the frequency and number of splenic iNKT cells in B-$Cd1d^{-/-}$ and B-WT mice that received α-GalCer or vehicle alone ($n = 3$ per group, one of two experiments is shown). **d** Representative histograms and bar chart showing MFI of Ki-67 in splenic iNKT cells from α-GalCer- or vehicle-treated B-$Cd1d^{-/-}$ and B-WT mice ($n = 3$ per group, one of two experiments is shown). **e** Representative histograms and bar chart showing MFI of PLZF in splenic iNKT cells from α-GalCer- or vehicle-treated B-$Cd1d^{-/-}$ and B-WT mice ($n = 3$ per group, one of two experiments is shown). **f** Representative flow cytometry plots and bar chart showing the frequency of IFN-γ$^+$PLZF$^+$ splenic iNKT cells from α-GalCer- or vehicle-treated B-$Cd1d^{-/-}$ and B-WT mice ($n = 4$ per group, one of two experiments is shown). **g** Representative histograms and bar charts showing MFI of T-bet and GATA3 in splenic iNKT cells from α-GalCer- or vehicle-treated B-$Cd1d^{-/-}$ and B-WT mice ($n = 3$ per group, one of two experiments is shown). **h** Representative immunofluorescence showing the localization of splenic NK1.1$^+$CD3$^+$ T cells and IgD$^+$ B cells in B-$Cd1d^{-/-}$ and B-WT chimeric mice that received α-GalCer or vehicle alone. Arrowheads point to NK1.1$^+$CD3$^+$ cells. Bar, 100 μm, inset bar, 5 μm ($n = 3$ per group, one of two experiments is shown). **i** Mean clinical score of B-$Il10^{-/-}$ chimeric mice (right) and B-WT chimeric mice (left) that received α-GalCer or vehicle alone following induction of arthritis; the Y axis shows the percentage of swelling in antigen-injected knee compared to control knee (B-$Il10^{-/-}$ ± α-GalCer $n = 5$ per group, B-WT ± α-GalCer $n = 6$ per group, one of two experiments is shown). **b** Analyzed on day 7 post disease onset, **c**, **d**, **h** analyzed on day 3 post disease onset, and **e–i** analyzed at 16 h post disease onset. Data are mean ± s.e.m. *$p < 0.05$, **$p < 0.01$, ***$p < 0.001$, and ****$p < 0.0001$ (**a**, **i** two-way ANOVA, **b–g** one-way ANOVA)

deficient mice (Fig. 6a, b). The disease progression in $Cd1d^{-/-}$ and WT mice is shown in Supplementary Fig 11a.

To confirm the involvement of CD1d in T2-MZP Breg-suppressive function and in the differentiation of disease-ameliorating iNKT cells, we isolated T2-MZP Bregs from the spleens of WT, $Ja18^{-/-}$, and $Cd1d^{-/-}$ mice in remission from arthritis and transferred them into WT mice. T2-MZP Bregs from WT and $Ja18^{-/-}$ mice (T2-MZP Bregs express CD1d in both strains) suppressed arthritis and significantly inhibited the frequency of IFN-$\gamma^+$CD4$^+$ T cells in WT recipient mice ($p <$

0.01 by one-way ANOVA) (Fig. 6c, d). In contrast, T2-MZP Bregs from $Cd1d^{-/-}$ mice failed to inhibit arthritis or to limit Th1/Th17 responses in the recipient WT mice (Fig. 6e, f). Transfer of WT and $Ja18^{-/-}$T2-MZP, but not $Cd1d^{-/-}$ T2-MZP B cells, caused an increase in iNKT cells producing IFN-$\gamma$ (Fig. 6g). Of note, we could not detect differences in IL-10 production by T2-MZP B cells isolated from $Cd1d^{-/-}$ and WT mice (Supplementary Fig. 11b). Our data demonstrate that suppression of arthritis by T2-MZP Bregs requires both iNKT cells and CD1d.

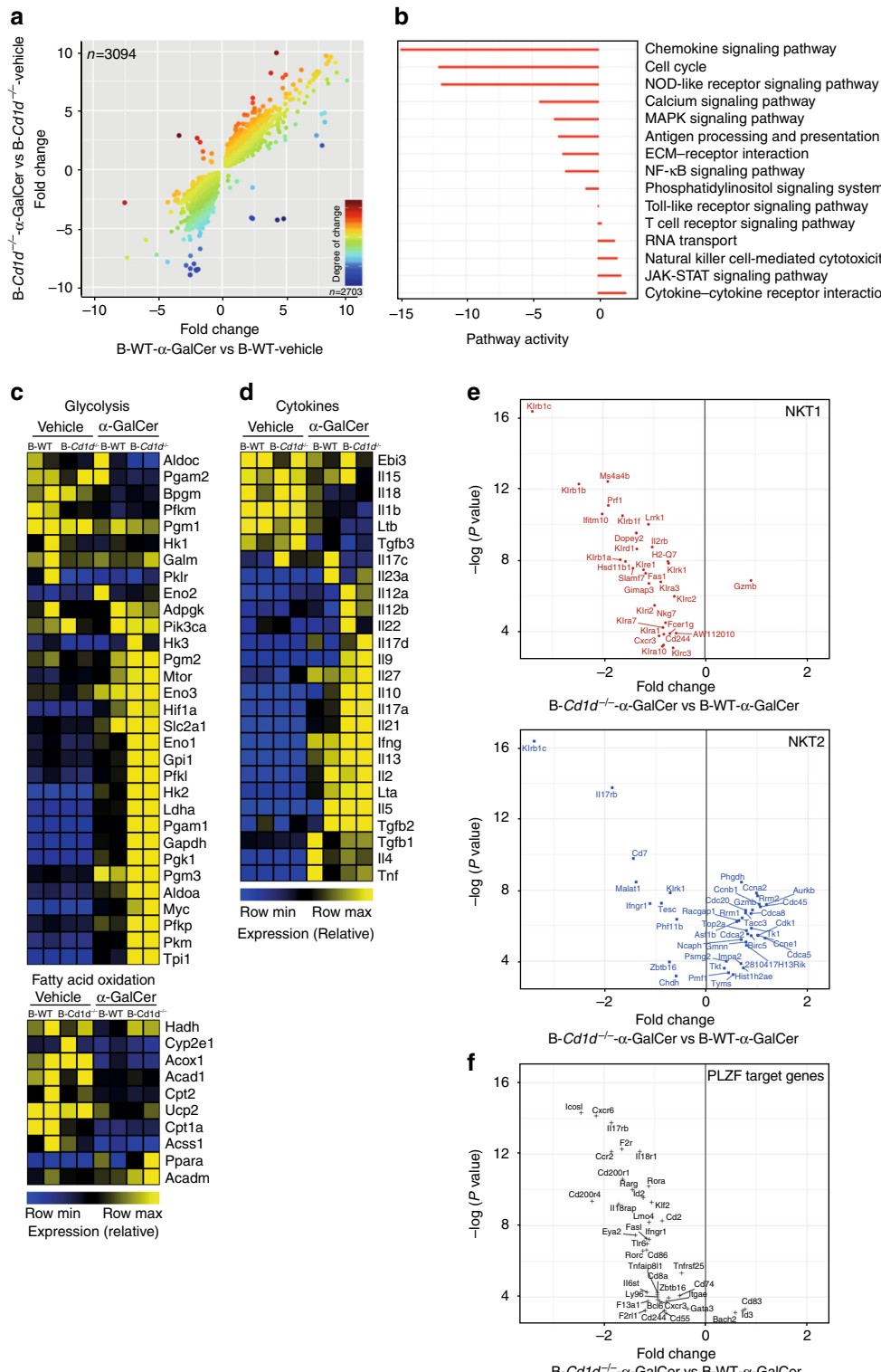

**Redundant role of MZ B cells in α-GalCer suppression**. MZ B cells also express high levels of CD1d (Supplementary Fig. 9a, c) and at least in vitro have been shown to exert suppression[42]. We have previously reported that adoptively transferred MZ B cells are unable to suppress arthritis in the recipient mice[39], likely due to their inability to repopulate the spleen. Next, we depleted MZ B cells in WT mice, by administering anti-CD11a and anti-CD49d antibodies[43,44] prior to the induction of arthritis and treatment with α-GalCer. Isotype-treated mice were used as controls. No changes in the number of T2-MZP B cells were observed in MZ B cell-depleted compared to control mice (Supplementary Fig. 12). α-GalCer significantly reduced the disease in both MZ B cell-depleted and control mice ($p < 0.05$ by two-way ANOVA) (Fig. 7a). The reduction in disease severity was accompanied by iNKT cell expansion and increased expression of Ki-67 (Fig. 7b, c). The induction of PLZF and IFN-γ expression in iNKT cells from both MZ B cell-depleted and control mice was also comparable (Fig. 7d, e).

**iNKT cells are dispensable for Breg development**. Recently, it has been shown that iNKT cells can induce the expansion of IL-10-producing Bregs in a cognate manner[45]. We therefore explored the existence of a feedback loop, whereby iNKT cells support the differentiation of IL-10-producing Bregs. We did not observe an enhancement of B cell IL-10 expression following administration of α-GalCer to IL-10-eGFP reporter mice (Fig. 8a). We induced arthritis in iNKT-cell-deficient $J\alpha18^{-/-}$ and $Cd1d^{-/-}$ mice to explore whether iNKT cells induced Bregs. As shown before, B cell development was comparable in iNKT-cell-deficient and WT mice (Supplementary Fig. 13)[32]. We could not find any differences in the frequency of IL-10⁺ B cells and the amount of IL-10 produced by B cells in response to different stimuli in $J\alpha18^{-/-}$, $Cd1d^{-/-}$, and WT mice (Fig. 8b, c). These results demonstrate that in this model iNKT cells are not required for the differentiation of Bregs.

## Discussion

Expression of CD1d is a hallmark of several described Breg phenotypes, including IL-10-producing B cells (B10), T2-MZP, and MZ B cells[2]. Here, we show that the CD1d-dependent Breg interaction with iNKT cells is critical for the differentiation of iNKT cells with suppressive capacity.

B cells have diverse immunological functions, including presentation of peptide and lipid antigens. While the upregulation of MHC class II molecules upon B cell activation promotes the activation of CD4⁺ helper T cells, CD1d-dependent presentation of α-GalCer by B cells to iNKT cells generates an activation signal that drives antibody production[18,19,46–48]. Emerging evidence shows that optimal CD1d expression on B cells is essential for the generation of iNKT cell responses. For example, the loss of CD1d expression on B cells, following Epstein–Barr virus infection, results in severe attenuation of iNKT cell functions[49]. Data from our laboratory and confirmed by other groups[1,50] suggested that aberrant CD1d expression by B cells is associated with iNKT cell functional defect observed in SLE patients. These findings suggest that CD1d-mediated B-iNKT cell cross talk may be important in iNKT cell homeostasis and for regulation of suppression.

Using a combination of chimeric mice and adoptive transfer strategies, we have demonstrated that CD1d⁺ Bregs inhibit inflammation by directly interacting with iNKT cells, thus underscoring a new mechanism for Breg suppression. It is becoming increasingly clear that Bregs can suppress in an IL-10-independent manner, including via IL-35, TGF-β, adenosine, or PD-L1[4,5,51]. In contrast to the induction of CD4⁺ Tregs by Bregs[52], which is impaired in the absence of IL-10⁺ B cells, here, we show that the differentiation of iNKT cells that limit inflammation in arthritis is CD1d-dependent but IL-10-independent. Indeed, arthritis in B-$Il10^{-/-}$ or in global $Il10ra^{-/-}$ mice was ameliorated by α-GalCer.

Several mechanisms for iNKT-cell-mediated regulation of pathogenic T helper cell responses that drive autoimmunity have been suggested[11–13]. For example, expression of IL-4, IL-10, and IFN-γ has been shown to mediate iNKT-cell-dependent inhibition of autoimmunity[13]. We found that iNKT cell IFN-γ upregulation was impaired in response to α-GalCer treatment in B cell-deficient or B-$Cd1d^{-/-}$ mice compared to control mice. The protective role of iNKT cell IFN-γ in arthritis was supported by the experiments where the neutralization of IFN-γ reversed the protective effect of α-GalCer and restored the subsequent proinflammatory cytokine production[24]. Grajewski et al. originally showed that iNKT cells producing IFN-γ ameliorate uveitis by inhibiting the adaptive Th1 and Th17 responses, suggesting that innate, iNKT-cell-produced IFN-γ is critical to initiate regulatory circuits that inhibit the differentiation of pathogenic T helper cells in uveitis. In EAE and type 1 diabetes, iNKT cells producing IFN-γ protect mice from developing the disease by dampening CD4⁺ T cell activation, including their production of IFN-γ and IL-17[25,27,53]. Of interest, at least in vitro iNKT cells isolated from SLE patients produce less IFN-γ compared to healthy individuals[1]. Although mice lacking B cells present a significant reduction in the frequency of IFN-γ⁺ iNKT cells compared to control group, in response to α-GalCer, few IFN-γ⁺ iNKT cells are induced. This raises the question of why the remaining iNKT cells fail to limit the severity of arthritis. It is tempting to speculate, also based on the results in Fig. 5e, which show that the maximum suppression of adaptive T cell responses occurs in the presence of high levels of IFN-γ, that there may be a dose effect of iNKT cells producing IFN-γ on the severity of arthritis. Therefore, we propose that early (innate) production of IFN-γ by iNKT cells is protective, but late (adaptive) production

**Fig. 4** B cell inability to present α-GalCer affects iNKT cell expression of genes involved in cytokine responses and metabolism. **a** Scatter plot showing fold-change relationships between common expressed genes ($n = 5797$) in iNKT cells from B-$Cd1d^{-/-}$ and B-WT mice that received α-GalCer or vehicle alone following induction of arthritis (data sets from RNA-seq). Color gradient indicates the degree of difference in fold-change values between iNKT cells in B-$Cd1d^{-/-}$ and B-WT mice following α-GalCer treatment. **b** Pathway enrichment analysis of pathways (right) showing enrichment among genes expressed differentially in iNKT cells from B-$Cd1d^{-/-}$ and B-WT mice following α-GalCer treatment. It is presented as net pathway perturbation in iNKT cells from B-$Cd1d^{-/-}$ mice compared to B-WT mice. Negative values mean inhibited in iNKT cells from B-$Cd1d^{-/-}$, positive values mean activated in iNKT cells from B-$Cd1d$. **c** Heat maps showing the expression of genes involved in glycolysis (top) and fatty acid oxidation (bottom) by iNKT cells from B-$Cd1d^{-/-}$ and B-WT mice that received α-GalCer or vehicle alone. **d** Heat map showing the expression of cytokine genes by iNKT cells from B-$Cd1d^{-/-}$ and B-WT mice that received α-GalCer or vehicle alone. **e, f** Volcano plots showing fold changes of differentially expressed genes associated with **e** NKT1 and NKT2 phenotypes, and **f** PLZF transcriptional regulation, between iNKT cells from B-$Cd1d^{-/-}$ and B-WT mice following α-GalCer treatment, plotted against $p$ values

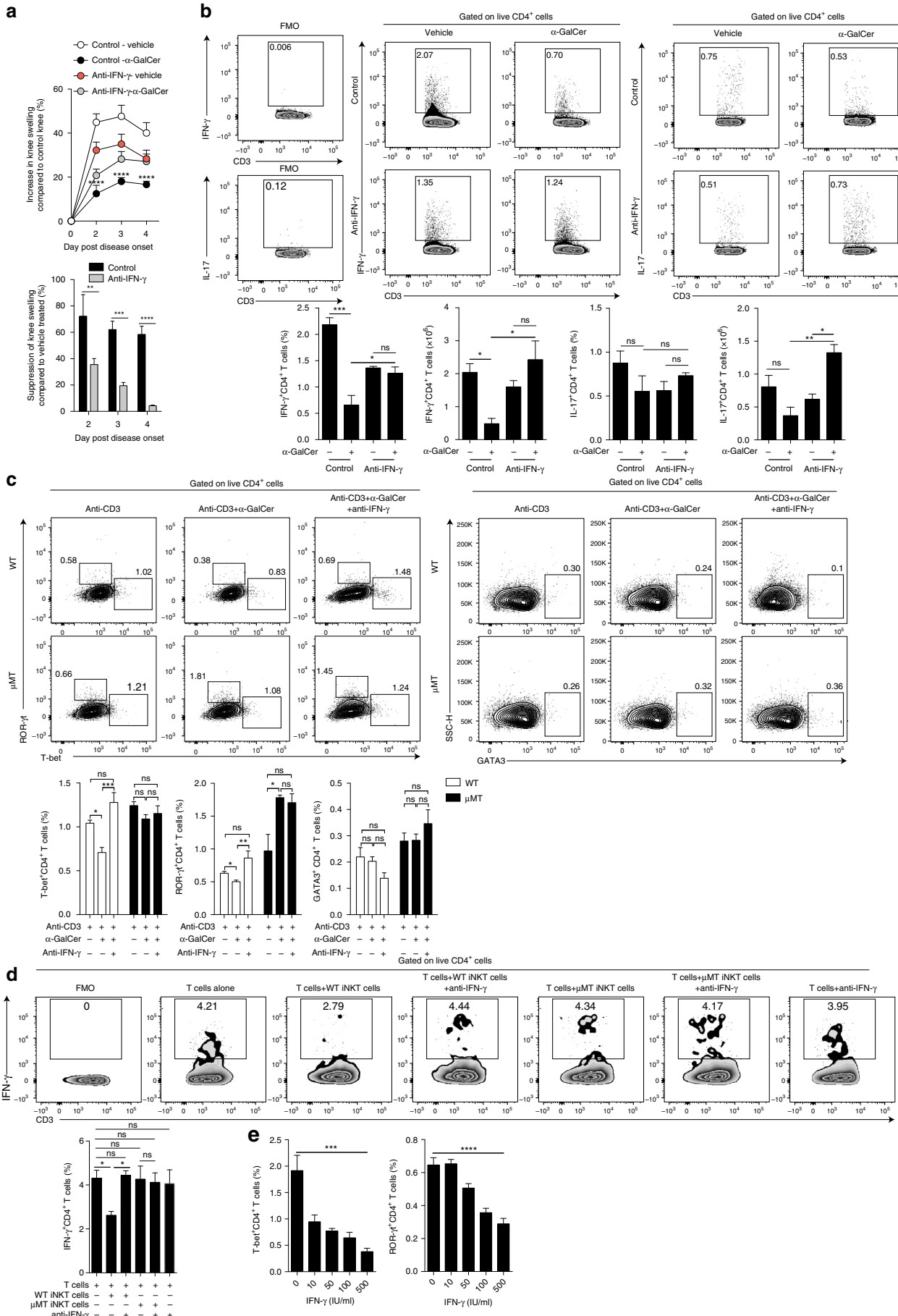

of IFN-γ (together with IL-17) by CD4[+] T cells is associated with increased inflammation and pathology. However, we cannot exclude that in addition to IFN-γ, CD1d[+] B cell-driven iNKT cells suppress via other mechanisms yet to be unraveled.

In agreement with our data, several reports using different experimental models of RA, including collagen-induced arthritis (CIA) and glucose-6-phosphate isomerase (GPI)-induced arthritis, have shown that α-GalCer ameliorates arthritis via inhibition of proinflammatory cytokine production by CD4[+] T cells[54,55]. Coppieters et al. also showed that high levels of IFN-γ accumulate in the serum of α-GalCer-treated mice and that neutralization of IFN-γ at the time of α-GalCer administration during disease onset reduced the efficacy of the treatment, which again supports our data showing that α-GalCer protection is mediated by IFN-γ[+] iNKT cells[54]. In contrast to our results, it has been reported that the protective role of α-GalCer in CIA was also mediated by an upregulation of IL-10 by T cells, and that administration of an anti-IL-10R antibody abrogated the protective effect[56]. In our model, suppression of arthritis following α-GalCer was IL-10 independent as there was equivalent suppression of the disease in α-GalCer-treated *Il10ra*[−/−] and WT mice. The same authors reported different effects of α-GalCer treatment on cytokine production between DBA/1 and C57BL/6 mice with CIA; higher levels of serum IFN-γ were observed in C57BL/6 compared to DBA/1 mice. This suggests that α-GalCer may mediate suppression differentially according to the disease model (CIA vs. AIA) and to the genetic background of the mice (C57BL/6 vs. DBA/1)[56].

The induction of different populations of regulatory iNKT cells, such as lymph node FoxP3[+] iNKT cells[28], adipose tissue-enriched FoxP3[−]IL-10[+] iNKT cells[57], or adipose tissue-resident FoxP3[−]PLZF[−]E4BP4[+] iNKT cells[29], has also been associated with iNKT cell anti-inflammatory effects. Our data suggest that Breg-induced iNKT cells are FoxP3[−]PLZF[+]E4BP4[−]. The different phenotype may reflect the tissue from which they were isolated, the particular inflammation model, or the stimuli received from Bregs.

The comparative analysis of gene expression in iNKT cells from B-*Cd1d*[−/−] and B-WT mice showed changes in several regulatory pathways. In particular, in the absence of CD1d-presenting B cells, we observed a downregulation of many genes that are enriched in NKT1 cells and the upregulation of those associated with NKT2 phenotype[36]. The expression of PLZF has been linked with the acquisition and regulation of multiple facets of iNKT cell phenotype, development, homeostasis, and function[22,23]. Here, we consistently found PLZF expressed at lower levels when B cells were unable to present α-GalCer.

Considering the central role of PLZF in establishing iNKT cell function during differentiation and its maintenance in the periphery, it is tempting to speculate that T2-MZP Bregs via CD1d, and possibly other factors, may play a role in PLZF expression or in the stabilization of PLZF locus. More detailed experiments including the analysis of proximal conserved noncoding sequences in the PLZF locus should be performed in the future to assess this question in more detail. Taken together with the results showing alteration in genes involved in immunometabolism in iNKT cells from B-*Cd1d*[−/−] compared to B-WT mice, our data suggest that Bregs expressing CD1d coordinate the induction of iNKT cell effector program.

Recently, work carried out by Vomhof-DeKrey et al. showed that 4-hydroxy-3-nitrophenyl-α-GalCer administration does not induce humoral memory, but instead drives the expansion of IL-10[+] Bregs[45]. This suggests that iNKT cells provide pivotal signals for Breg differentiation in vivo. In contrast to these data, we show that Breg development is unaltered in mice lacking iNKT cells and we could not observe an induction of IL-10[+] B cells upon α-GalCer treatment. iNKT cells isolated from α-GalCer-treated AIA-immunized WT mice transferred disease suppression to μMT mice (Supplementary Fig. 14) and Bregs failed to suppress the disease in iNKT-cell-deficient mice. Therefore, at least in our experimental system, there is a "long-lasting" effect of B cell-primed iNKT cells that persists following the transfer of iNKT cells to a B cell-deficient host.

Given that α-GalCer presentation by B cells to iNKT cells is known to result in the differentiation of antibody-producing B cells, we also investigated whether plasma cells were induced in our model and whether they could regulate via the production of IL-35. Following α-GalCer administration, we did not observe induction of CD138[+] plasmablasts (Supplementary Fig. 15a), or any changes in *Ebi3* or *p35* mRNA levels by qPCR in B cells (Supplementary Fig. 15b, c).

The lack of involvement of IL-35-producing plasmablasts in this model was not entirely surprising as we have previously published that mice lacking the p35 subunit of IL-35 develop milder AIA compared to WT mice[58]. The decrease in disease severity was mirrored by an increase in T2-MZP Bregs and a decrease in MZ B cells. We have also shown that T2-MZP or B cells stimulated with anti-CD40, isolated from p35-deficient mice protect against AIA upon adoptive transfer to recipient mice. Therefore, in this system, IL-35 does not seem to play a regulatory function.

CD1d is associated with a Breg phenotype. In a spontaneous model of colitis, B cells isolated from CD1d-deficient mice failed to suppress the disease when transferred to TCRα[−/−] mice,

---

**Fig. 5** α-GalCer-mediated amelioration of arthritis is partially dependent on IFN-γ production by iNKT cells. **a** Top, mean clinical score of anti-IFN-γ-treated and isotype-control-treated WT mice that received α-GalCer or vehicle alone following induction of arthritis. The *Y* axis shows the percentage of swelling in antigen-injected knee compared to control knee. Bottom, bar chart showing the suppression of knee swelling in anti-IFN-γ-treated and isotype-control-treated WT mice that received α-GalCer, compared to anti-IFN-γ-treated and isotype-control-treated WT mice that received vehicle alone (anti-IFN-γ-treated WT ± α-GalCer *n* = 5 per group, isotype-control-treated WT ± α-GalCer *n* = 4 per group, one of two experiments is shown). **b** Representative flow cytometry plots (top) and bar charts (bottom) showing the frequency and number of splenic IFN-γ[+] and IL-17[+] CD4[+] T cells in anti-IFN-γ-treated and isotype-control-treated WT mice that received α-GalCer or vehicle alone (*n* = 3 per group, one of two experiments is shown). **c** Representative flow cytometry plots (top) and bar charts (bottom) showing the frequency of T-bet[+], ROR-γt[+], and GATA3[+] CD4[+] T cells in μMT and WT splenocyte cultures with α-GalCer and isotype control or α-GalCer and anti-IFN-γ-blocking antibodies (μMT *n* = 3, WT *n* = 6, one of two experiments is shown). **d** Representative flow cytometry plots (top) and bar chart (bottom) showing the frequency of IFN-γ[+]CD4[+] T cells following coculture with in vivo α-GalCer-stimulated iNKT cells (or alone), isolated from the spleens of arthritic μMT and WT mice in the presence of IFN-γ-neutralizing or isotype-control antibodies (*n* = 4 per condition, one of two experiments is shown). **e** Bar charts showing the frequency of T-bet[+] (left) and ROR-γt[+] (right) CD4[+] T cells, isolated from the spleens of arthritic WT mice, following stimulation with increasing concentrations of IFN-γ (*n* = 3 per condition, one of two experiments is shown). **b** Analyzed on day 3 post disease onset. Data are mean ± s.e.m. ns not significant, *$p < 0.05$, **$p < 0.01$, ***$p < 0.001$, and ****$p < 0.0001$ (**a** two-way ANOVA, **b**–**e** one-way ANOVA)

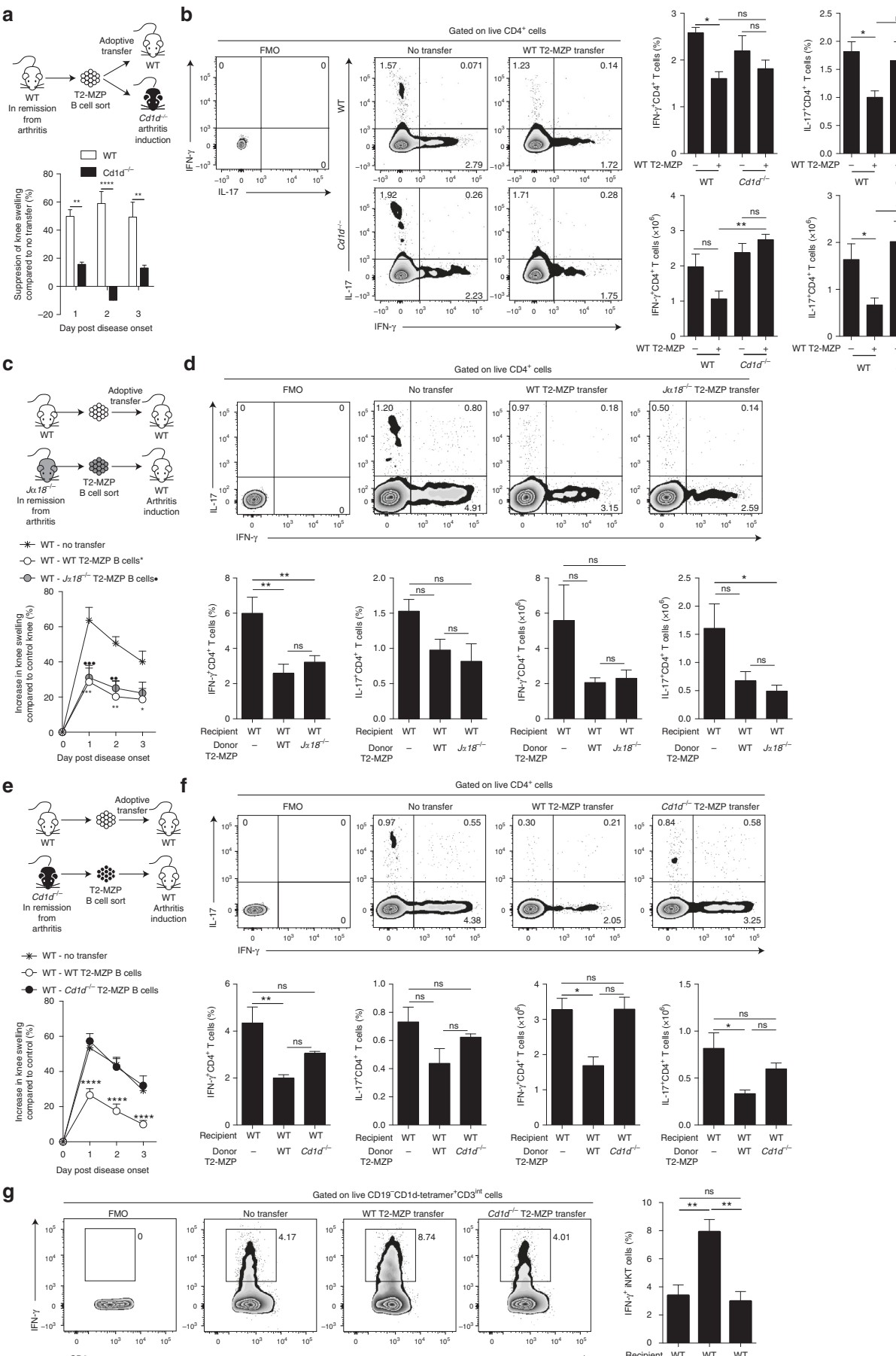

confirming the importance of CD1d in the regulatory function of B cells. In colitis, Breg-mediated suppression required IL-10 and not iNKT cells, as B cells could still suppress in the absence of iNKT cells. Similarly, Bregs, induced in response to *S. mansoni*, suppress allergic airway inflammation in mice in a CD1d-dependent, but iNKT-cell-independent manner[7]. Differences between our results and those previously published could be attributed to the fact that while AIA is a Th1-driven model, both colitis in TCRα[−/−] mice and *S. mansoni*-induced allergic airway inflammation are Th2 driven. It is conceivable that B cells expressing CD1d may inhibit Th1 and Th2 cell responses differently (e.g., via iNKT cells in Th1- and via IL-10 in Th2-driven models). In addition, activation of epithelial cells using an anti-CD1d antibody has been shown to induce the upregulation of IL-10 in the epithelial cells[59]. Thus, direct activation of CD1d-expressing cells, in the absence of iTCR ligation, could provide an additional explanation for the CD1d-dependent but iNKT-cell-independent and IL-10-mediated effects reported by others[59].

Previous work has shown that in an in vivo killing assay, α-GalCer-loaded, CD1d-expressing cells, in particular MZ B cells and to a certain extent CD11c[+] and CD11b[+] cells are killed[68]. In arthritis, we found that splenic B cell subset frequencies were not reduced and the numbers of each subset were increased in α-GalCer-treated B-WT mice. In α-GalCer-treated B-*Cd1d*[−/−] mice, MZ B cell frequency was reduced (Supplementary Fig. 16). MZ B cells express higher levels of CD1d than T2-MZP B cells; however, our results show that MZ B cell depletion does not impair the induction of suppressive iNKT cells following α-GalCer treatment. We have also shown that there were 50% less IFN-γ[+] iNKT cells in the liver compared to the spleen after α-GalCer treatment of arthritic mice. T2-MZP B cells do not reside in the liver (Supplementary Fig. 17) and in this organ, the majority of B cells are follicular which express low levels of CD1d, providing indirect evidence that T2-MZP Bregs are important for the differentiation of suppressive IFN-γ[+] iNKT cells. One possible explanation of how MZ B cells contribute differentially to iNKT-cell-driven disease outcome is the location. To exclude a role for MZ B cells, the effect of α-GalCer in the absence of T2-MZP B cells should be assessed. However, this is technically impossible, thus leaving open the possibility that MZ B cells may be important for the α-GalCer-mediated suppression.

In conclusion, we have described a T2-MZP Breg regulatory mechanism that acts as a rheostat in the differentiation of iNKT cells with suppressive capacity. In an era where B cells constitute a primary therapeutic target in several autoimmune diseases and in cancer, these data add a further degree of complexity that needs to be considered when designing new therapies aimed to eliminate the entirety of B cells.

## Methods

**Mice.** C57BL/6 mice were from Harlan Laboratories (now Envigo Laboratories). μMT[60] (stock no: 002288), *Cd1d*[−/−][41] (stock no: 008881), and *Il10*[−/−][61] (stock no: 002251) mice were from The Jackson Laboratory. CD11c-DTR mice[62] were provided by C. Bennett, University College London, UK and were originally from The Jackson Laboratory (stock no: 004509). *Jα18*[−/−] mice[40] were provided by A. Karadimitris, Imperial College London, UK. *Il10ra*[−/−] mice[63] were provided by W. Muller, University of Manchester, UK. IL-10 transcriptional reporter (Vert-X) mice[64] were provided by C. Karp, Bill & Melinda Gates Foundation. All mice were on C57BL/6 background. Mice were age- and sex-matched (males), and used at 6–12 weeks of age. Mice were bred and maintained under specific pathogen-free (SPF) conditions at the animal facility at University College London. All experiments were approved by the Animal Welfare and Ethical Review Body of University College London and authorized by the United Kingdom Home Office.

**Induction of antigen-induced arthritis.** AIA was induced[52]. Male mice were immunized subcutaneously at the tail base with 200 μg of methylated bovine serum albumin (mBSA, Sigma-Aldrich) in 200 μl of complete Freund's adjuvant (CFA) and phosphate-buffered saline (PBS, Sigma-Aldrich). A concentration of 3 mg/ml CFA was prepared by mixing of *Mycobacterium tuberculosis* (Difco) in incomplete Freund's adjuvant (IFA, Sigma-Aldrich). After 7 days, mice received an intra-articular injection of 200 μg of mBSA in 10 μl of PBS in the right knee or PBS alone as a control in the left knee. Joint size was measured using POCO 2-T calipers (Kroeplin Längenmesstechnik), and swelling was calculated as percentage increase in size between antigen- and control-injected knee.

**Administration of α-galactosylceramide.** When treated with α-galactosylceramide (α-GalCer, Enzo Life Sciences), mice received an intraperitoneal injection of 2 μg of α-GalCer in 200 μl of 0.5% DMSO (Sigma-Aldrich) in PBS at the time of the intra-articular injection, unless otherwise indicated. As a control, a separate group of mice was treated with 200 μl of 0.5% DMSO in PBS.

**B cell depletion.** For B cell depletion, mice were treated intraperitoneally with 200 μg of anti-CD20 antibodies (5D2, Genentech) in 200 μl of PBS, or isotype-control antibodies (IgG2a, 400502, Biolegend) in PBS, on days −21, −14, and −7 prior to intra-articular injection to induce AIA.

**CD11c[+] cell depletion.** CD11c-DTR mice were treated intraperitoneally with 100 ng of diphtheria toxin from *Corynebacterium diphtheriae* (Sigma-Aldrich) in 200 μl of PBS, or PBS alone as control[31] at the time of intra-articular injection to induce AIA, and 8 h prior to α-GalCer or vehicle administration.

---

**Fig. 6** T2-MZP Bregs mediate suppression via iNKT cells and require CD1d. **a** Top, schematic showing the experimental design. Bottom, bar chart showing suppression of knee swelling in *Cd1d*[−/−] and WT mice that received T2-MZP B cells, isolated from the spleens of WT mice on day 7 post disease onset, compared to *Cd1d*[−/−] and WT mice that received no transfer (*Cd1d*[−/−] *n* = 11, WT *n* = 16, one of two experiments is shown). **b** Representative flow cytometry plots (left) and bar charts (right) showing the frequency and number of splenic IFN-γ[+] and IL-17[+] CD4[+] T cells in *Cd1d*[−/−] and WT mice that received T2-MZP B cells, isolated from the spleens of WT mice on day 7 post disease onset, or *Cd1d*[−/−] and WT mice that received no transfer (*n* = 3 per group, one of two experiments is shown). **c** Top, schematic showing the experimental design. Bottom, mean clinical score of WT mice that received T2-MZP B cells, isolated from the spleens of *Jα18*[−/−] or WT mice on day 7 post disease onset, or WT mice that received no transfer following induction of arthritis. The *Y* axis shows the percentage of swelling in antigen-injected knee compared to control knee (*Jα18*[−/−] T2-MZP transfer *n* = 4, WT T2-MZP transfer *n* = 3, no transfer *n* = 4, one of two experiments is shown). **d** Representative flow cytometry plots (top) and bar charts (bottom) showing the frequency and number of splenic IFN-γ[+] and IL-17[+]CD4[+] T cells in WT mice that received T2-MZP B cells, isolated from the spleens of *Jα18*[−/−] or WT mice on day 7 post disease onset, or WT mice that received no transfer (*n* = 4 per group, one of two experiments is shown). **e** Top, schematic showing the experimental design. Bottom, mean clinical score of WT mice that received T2-MZP B cells, isolated from the spleens of *Cd1d*[−/−] or WT mice on day 7 post disease onset, or WT mice that received no transfer following induction of arthritis. The *Y* axis shows the percentage of swelling in antigen-injected knee compared to control knee (*n* = 6 per group, one of two experiments is shown). **f** Representative flow cytometry plots (top) and bar charts (bottom) showing the frequency and number of splenic IFN-γ[+] and IL-17[+]CD4[+] T cells in WT mice that received T2-MZP B cells, isolated from the spleens of *Cd1d*[−/−] or WT mice on day 7 post disease onset, or WT mice that received no transfer (*n* = 4 per group, one of two experiments is shown). **g** Representative flow cytometry plots and bar chart showing the frequency of IFN-γ[+] splenic iNKT cells in WT mice that received T2-MZP B cells, isolated from the spleens of *Cd1d*[−/−] or WT mice on day 7 post disease onset, or WT mice that received no transfer (*n* = 4 per group, one of two experiments is shown). **b, d, f, g** Analyzed on day 3 post disease onset. Data are mean ± s.e.m. ns not significant, *$p < 0.05$, **$p < 0.01$, ***$p < 0.001$, ****$p < 0.0001$, ••$p < 0.01$, and •••$p < 0.001$ (**a, c, e** two-way ANOVA; **b, d, f** one-way ANOVA)

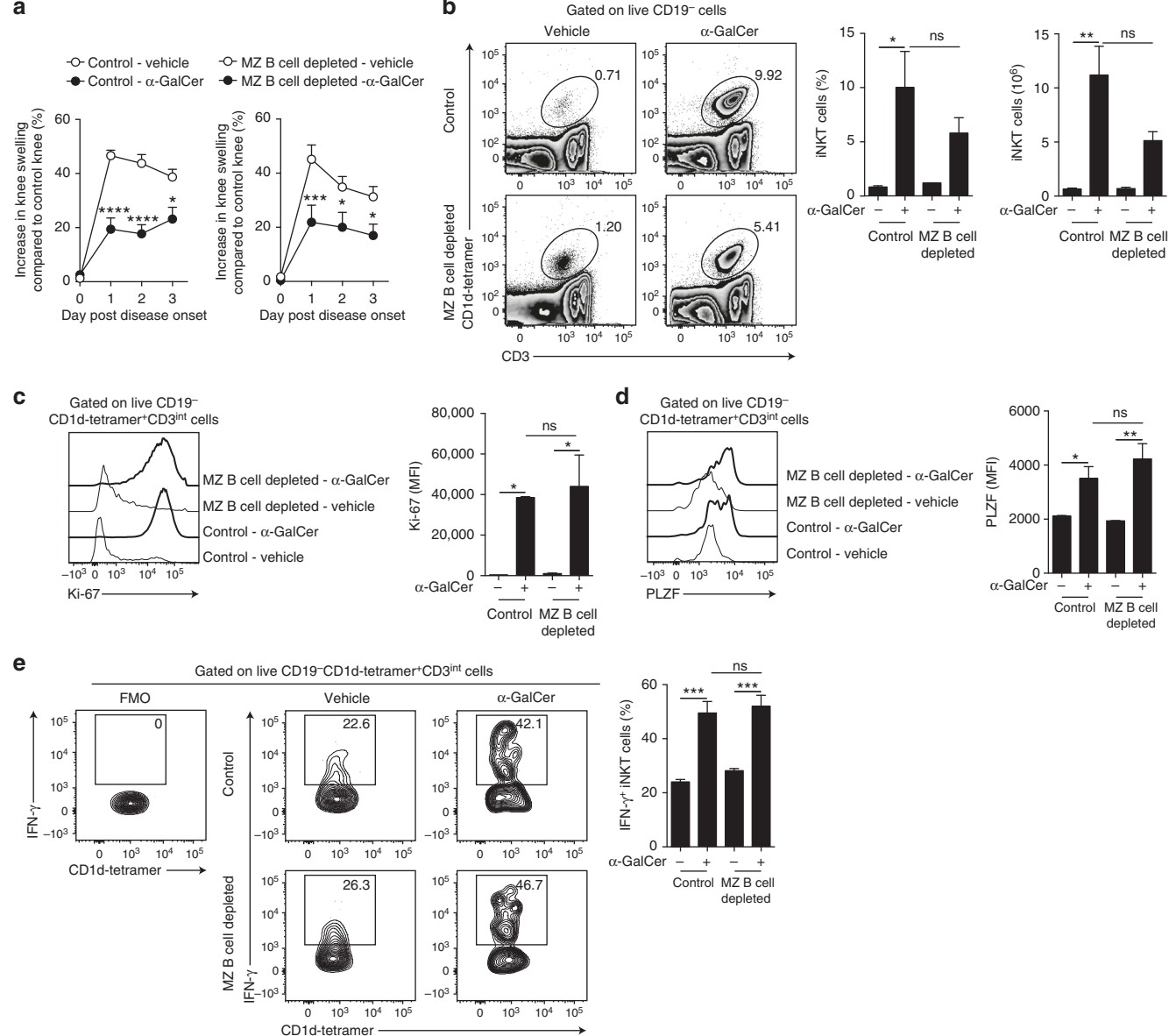

**Fig. 7** MZ B cells are dispensable for α-GalCer-mediated amelioration of arthritis. **a** Mean clinical score of control mice (left) or MZ B cell-depleted mice (right) that received α-GalCer or vehicle alone following induction of arthritis. The Y axis shows the percentage of swelling in antigen-injected knee compared to control knee (α-GalCer-treated $n = 5$ per group, vehicle-treated $n = 6$ per group, one of two experiments is shown). **b** Representative flow cytometry plots and bar charts showing the frequency and number of splenic iNKT cells in MZ B cell-depleted mice and control mice that received α-GalCer or vehicle alone ($n = 3$ per group, one of two experiments is shown). **c** Representative histograms and bar chart showing MFI of Ki-67 in splenic iNKT cells from α-GalCer- or vehicle-treated MZ B cell-depleted mice and control mice ($n = 3$ per group, one of two experiments is shown). **d** Representative histograms and bar chart showing MFI of PLZF in splenic iNKT cells from α-GalCer- or vehicle-treated MZ B cell-depleted mice and control mice ($n = 3$ per group, one of two experiments is shown). **e** Representative flow cytometry plots and bar chart showing the frequency of IFN-γ$^+$ splenic iNKT cells from α-GalCer- or vehicle-treated MZ B cell-depleted mice and control mice ($n = 4$ per group, one of two experiments is shown). **b**, **c** Analyzed on day 3 post disease onset; **d**, **e** analyzed at 16 h post disease onset. Data are mean ± s.e.m. ns not significant, $^*p < 0.05$, $^{**}p < 0.01$, $^{***}p < 0.001$, and $^{****}p < 0.0001$ (**a** two-way ANOVA; **b**–**e** one-way ANOVA)

**Marginal zone B cell depletion**. For marginal zone B cell depletion, mice were treated intraperitoneally with 100 μg of each anti-CD11a (M17/4, eBioscience) and anti-CD49d (cR1-2, Biolegend), or isotype-control antibodies (IgG2a, 400502, Biolegend), 16 h prior to intra-articular injection to induce AIA.

**In vivo IFN-γ neutralization**. For IFN-γ neutralization, mice were treated intraperitoneally with 200 μg of anti-IFN-γ antibodies (XMG1.2, BioXCell) in 200 μl of PBS, or isotype-control antibodies (IgG1, 400402, Biolegend) in PBS 2 days prior to intra-articular injection, and at day 0 of and 2 days post intra-articular injection with mBSA.

**Cell preparation**. Spleens and livers were collected in complete RPMI-1640 with L-glutamine and NaHCO₃ (cRPMI, Sigma-Aldrich) with 1:10 FBS (Biosera), 1:100 penicillin (10,000 units/ml)/streptomycin (10 mg/ml) (Sigma-Aldrich), and 1:1000 50 mM 2-mercaptoethanol (Gibco, Invitrogen). Single-cell suspensions were obtained by dissociating the tissue through 70-μM cell strainers (BD Biosciences), and erythrocytes in spleens were lysed using red blood cell (RBC) lysis buffer (Sigma-Aldrich). For the isolation of liver mononuclear cells, Percoll (Amersham Biosciences) 30%/70% gradient was used with subsequent RBC lysis.

**Generation of mixed bone marrow chimeric mice**. Mixed BM chimeric mice were generated. Briefly, recipient WT mice received 8 Gy of γ-irradiation via a

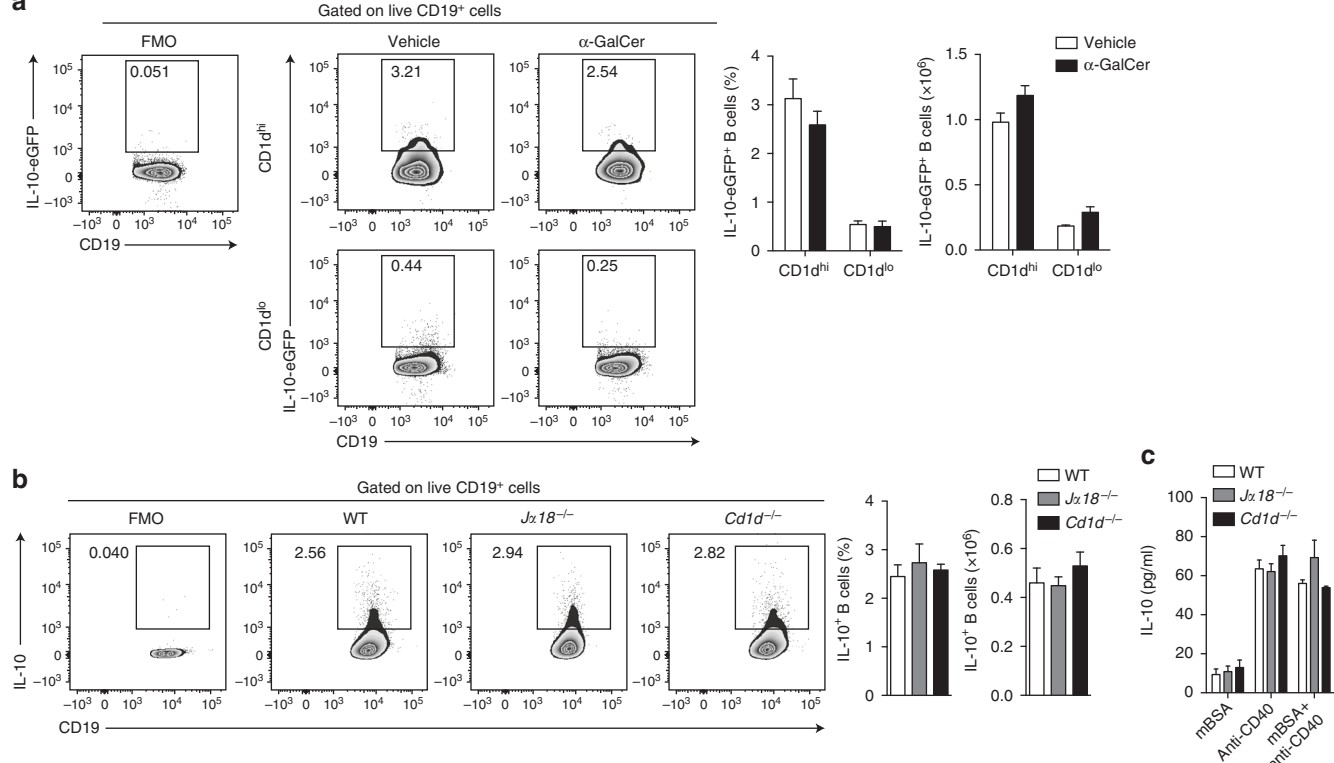

**Fig. 8** Regulatory B cell development and in vitro suppressive capacity are not impaired in iNKT-cell-deficient mice. **a** Representative flow cytometry plots and bar charts showing the frequency and number of splenic IL-10-eGFP+CD19+ B cells from IL-10-eGFP mice that received α-GalCer or vehicle alone following induction of arthritis (α-GalCer-treated group $n = 4$, vehicle-treated group $n = 3$, one of two experiments is shown). **b** Representative flow cytometry plots and bar charts showing the frequency and number of splenic IL-10+CD19+ B cells in $Ja18^{-/-}$, $Cd1d^{-/-}$, and WT mice following arthritis ($Ja18^{-/-}$ $n = 5$, $Cd1d^{-/-}$ $n = 8$, and WT $n = 9$, one of two experiments is shown). **c** Bar chart showing IL-10 production by B cells isolated from the spleens of arthritic $Ja18^{-/-}$, $Cd1d^{-/-}$, and WT mice following stimulation with methylated BSA (mBSA), anti-CD40, or mBSA+anti-CD40 as measured by ELISA ($n = 9$ per group, one of three experiments is shown). **a–c** Cells isolated on day 7 post disease. Data are mean ± s.e.m. (**a**, **c** two-way ANOVA; **b** one-way ANOVA)

cesium ($^{137}$Cs) source. Recipient mice were allowed to equilibrate for 5 h before administering $10^7$ donor BM cells. To restrict genetic deficiency of CD1d or IL-10 to B cells, lethally irradiated (8 Gy) WT mice were reconstituted with a mixture consisting of 80% BM from μMT mice and 20% BM from $Cd1d^{-/-}$ or $Il10^{-/-}$ mice. Control mice received 80% BM from μMT mice and 20% BM from WT mice. The efficiency of irradiation of WT mice was confirmed by the absence of B cells in mice reconstituted with 100% μMT BM. Chimeric mice were left to reconstitute for at least 8 weeks prior to use in AIA experiments.

**Adoptive transfer**. T2-MZP B cells were isolated from the spleens of $Ja18^{-/-}$, $Cd1d^{-/-}$, or WT mice on day 7 post disease onset by fluorescence-activated cell sorting (FACS). iNKT cells were isolated from the spleens of WT mice on day 3 following α-GalCer treatment by FACS. A total of $2 \times 10^6$ T2-MZP B cells or iNKT cells were then transferred intravenously into $Cd1d^{-/-}$ and WT, or μMT and WT mice, respectively, immediately following induction of arthritis. The control group received a Hanks' Balanced Salt Solution (Sigma-Aldrich) injection.

**Flow cytometry and fluorescence-activated cell sorting**. Single-cell suspensions were incubated with FcR-blocking reagent for the mouse (Miltenyi Biotec) prior to staining with specific antibodies (Supplementary Table 1) and CD1d tetramers. CD1d tetramers were conjugated to PE, AF488, APC, and AF647 and loaded with PBS57 or left unloaded, and were obtained from the National Institutes of Health Tetramer Core Facility. Tetramer staining was carried out prior to surface antibody staining. All staining was performed at 4 °C for 30 min. For transcription factor and Ki-67 staining, cells were fixed and permeabilized using Foxp3/Transcription Factor Staining Buffer Set (00-5523-00, eBioscience) according to the manufacturer's instructions. For the detection of IFN-γ and IL-17, cells were stimulated for 5 h in cRPMI with 50 ng/ml phorbol 12-myristate 13-acetate (PMA) and 250 ng/ml ionomycin (both from Sigma-Aldrich). For the detection of IL-10, cells were stimulated for 5 h in cRPMI with 100 ng/ml PMA and 1000 ng/ml ionomycin. Protein transport inhibitor (either containing brefeldin A or monensin, both from BD Biosciences) was added during stimulations according to the manufacturer's instructions. Cells were then stained for surface markers, fixed, and permeabilized

using Intracellular Fixation & Permeabilization Buffer Set (88-8824-00, eBioscience) according to the manufacturer's instructions. Dead cells were excluded with LIVE/DEAD Fixable Dead Cell Stains or DAPI (both from Invitrogen). Flow cytometry was performed on LSRFortessa, LSRFortessa X-20, LSRII, and cell sorting on FACSAria cell sorter (all from BD Biosciences). Data were analyzed using FlowJo (Treestar).

**Immunofluorescence**. Spleens were dissected, embedded in OCT compound (Tissue-Tek), and snap-frozen for cryosectioning (6 μm). Slides were then incubated in 100% ethanol to fix for 5–10 min (4 °C), followed by rehydration in PBS for 5 min (4 °C). The sections were blocked with 10% normal goat serum for 20 min at room temperature (RT), and then rinsed with PBS. The tissues were incubated with primary antibodies for 1 h at RT and then washed 3× in PBS (2 min for each wash). When secondary antibodies were used, the tissue was further incubated with the secondary antibody for 1 h at RT, again followed by three washes in PBS (2 min for each wash). The slides were finally mounted in Prolong Antifade Mountant with DAPI (Thermo Fisher), imaged on a Leica SPE confocal microscope using LAS X software, and analyzed using Fiji (ImageJ).

**In vitro B cell culture**. Splenic B cells were negatively purified (>95%) by magnetic separation using CD43 (Ly-48) microbeads (Miltenyi Biotec) and cultured for 72 h with 10 μg/ml mBSA, 10 μg/ml anti-CD40 (FGK4.5, BioXCell), or mBSA+anti-CD40. Supernatants from cell cultures were harvested and IL-10 was measured using mouse IL-10 DuoSet ELISA (DY417, R&D Systems) according to the manufacturer's instructions.

**In vitro iNKT cell suppression assay**. Splenic iNKT cells from arthritic μMT and WT mice were sorted by FACS 16 h following α-GalCer treatment. iNKT cells were then cocultured at a ratio of 1:1 with CD4+CD25− T cells, isolated from the spleens of WT mice 16 h following intra-articular injection to induce arthritis by magnetic separation (Miltenyi Biotec), for 72 h in the presence of 0.5 μg/ml plate-bound anti-CD3 (145-2C11, BD Biosciences). PMA, ionomycin, and protein transport

inhibitor were added for the last 6 h of culture. Following stimulation, cells were analyzed for the expression of IFN-γ by flow cytometry.

**In vitro B cell and dendritic cell depletion**. Splenocytes were negatively depleted of B cells (CD19) or dendritic cells (CD11c) by FACS and stimulated with IL-2 (80 U/ml, Miltenyi Biotec) in the presence of 100 ng/ml α-GalCer for 48 h.

**RNA sequencing**. Splenic iNKT cells from B-$Cd1d^{-/-}$ and B-WT mice that received α-GalCer or vehicle as control were sorted by FACS. Sorted cells were >95% pure. Total RNA was isolated using PicoPure RNA Isolation Kit (KIT0204, Applied Biosystems) according to the manufacturer's instructions. cDNA was synthesized and amplified using SMARTer v4 Ultra Low Input Kit (634888, Clontech). A total of 200 pg of amplified cDNA was then converted to a sequencing library using the Nextera XT Kit (FC-131-1024, Illumina, 12 cycles of PCR). Samples were individually indexed during library preparation, which allowed pooling and sequencing on the Illumina NextSeq 500 platform (43-bp paired end). An average of 18M read pairs was obtained for each sample. Reads were adapter trimmed (FASTX-Toolkit) and aligned to the *Mus musculus* genome (mm10) using Tophat v2.0.7.

**Bioinformatics analysis of RNA-seq data**. Differential expression analysis was performed using the edgeR algorithm[65] under default settings. Genes with false-discovery rate (FDR) values <0.05 were considered to be differentially expressed. Significantly overrepresented pathways were identified using signaling pathway impact analysis (SPIA)[66]. Kyoto Encyclopedia of Genes and Genomes (KEGG) pathways database[67] was used as a reference, while full mouse genome was used as a background for enrichment.

**Statistical analysis**. All data are expressed as mean ± s.e.m. Replicates were biological replicates. Clinical scoring was performed blinded. Mice were assigned to treatment groups at random for all mouse studies and, where possible, mixed among cages. Careful consideration has been given to calculate the number of mice needed to give statistically significant results, while using the minimum number of mice possible. Numbers of mice for each experiment were decided after consultation with a statistician at UCL. For in vivo studies, power calculations were based on data showing mean maximum wild-type arthritic knee swelling of 2 mm with an s.d. of 0.39 mm, and an expected test group arthritic knee swelling of ±0.6 mm compared to control group depending on the experiment. Group sizes of 5 or above were sufficient to reach a statistical power of at least 80%. Further, experiments were performed independently in triplicate to control for experimental variation. Groups were compared with two-tailed Student's *t* test, or one- or two-way analysis of variance (ANOVA) followed by Bonferroni posttest with Prism (GraphPad). Data followed Gaussian distribution with similar variance between groups that were compared.

**Data availability**. The RNA-seq data have been deposited in ArrayExpress under E-MTAB-6381. The authors declare that the data supporting the findings of this study are available within the article and its supplementary information files, or are available upon reasonable requests to the authors.

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

## Acknowledgements

We like to thank Dr Claire Bennett for kindly providing the CD11c-DTR mice and advice on the animal model, Dr Chris Karp (Bill and Melinda Gates foundation) for providing the Vert-X mice, and the staff of the University College London Biological Services Unit for all the help with animal husbandry, Jamie Evans for cell sorting, Dr Madhvi Menon, Dr Paul Blair, Dr Scott Thomson, and Christopher J. M. Piper for critically reviewing the manuscript. This work was funded by UK Medical Research Council studentship G0900950 to K.O., European Community's Seventh Framework Programme (FP7-2007–2013) under grant agreement no. HEALTH-F2-2013-602114 (Athero-B-Cell) to C.M. and I.D., Arthritis Research UK Foundation fellowship 21141 to E.C.R., Wellcome Trust Intermediate Clinical Fellowship 097259/Z/11/Z to K.N. and C.M, Arthritis Research UK studentship 21257 to D.E.M., and Nuffield Foundation Oliver Bird Rheumatism Program studentship to A.B. All research at Great Ormond Street Hospital NHS Foundation Trust and UCL Great Ormond Street Institute of Child Health is made possible by the NIHR Great Ormond Street Hospital Biomedical Research Centre. The views expressed are those of the author(s) and not necessarily those of the NHS, the NIHR or the Department of Health.

## Author contributions

K.O. conceived the study, performed the experiments, analyzed and interpreted the data, and wrote the manuscript; E.C.R. performed the experiments and provided a critical review of the manuscript; A.B., D.E.M., and K.N. performed the experiments; I.D. performed the bioinformatics analysis of RNA-seq data; and C.M. conceived the study, analyzed and interpreted the data, and wrote the manuscript.

## Additional information

**Competing interests:** The authors declare no competing financial interests.

