## [Peer Review File · Nature Communications]

Reviewers' comments:

Reviewer #1 (NKT, CD1d, Breg)(Remarks to the Author):

This manuscript has investigated interactions between CD1d-expressing regulatory B cells and iNKT cells. Using a rheumatoid arthritis model in mice, the authors show that CD1d-mediated antigen presentation by Bregs is critical for iNKT cell function. These interactions were required for the suppressive effects of iNKT cell activation on arthritis. These effects were independent of IL-10 production by Breg cells. From these findings, the authors conclude a critical role of CD1d on regulatory B cells for regulating iNKT cell function, in a manner that is independent of IL-10 production by B cells.

General comments:

The manuscript includes an impressive amount of data, with a variety of complementary approaches such as cell depletion, knockout mice, mixed bone marrow chimeras, adoptive transfers, etc. The data are convincing, well-described, and potentially important for understanding interactions between regulatory B cells and iNKT cells during autoimmunity and inflammatory responses.

Minor comments:

1. Mizoguchi et al. (*Immunity* 2002; 16:219-230) were the first to provide evidence for a role of CD1d on B cells in controlling inflammatory responses in a colitis model. In this system, CD1d and IL-10 were important, but not NKT cells. This report should be included and differences with the present work may be briefly discussed. For example, in some cases signaling through CD1d (e.g., via antibody stimulation) has been shown to influence functions of CD1d-expressing cells, providing one potential explanation for CD1d-dependent but NKT cell-independent effects (I believe the Mizoguchi paper largely used B cells purified based on CD1d expression, presumably using mAbs that could have potentially activated the B cells).

2. A variety of prior studies have studied the role of iNKT cells and iNKT cell activation on different models of rheumatoid arthritis. This includes Coppieters et al. (*J. Immunol.* 2007; 179:2300-2309); Horikoshi et al. (*PLoSOne* 2012; e51215); Miellot et al. (*EJI* 2005; 35:3704-3713), and several others. Although the model employed here is different than most of the models employed in these prior studies, these studies should be briefly discussed, because the studies reported here might not apply to other experimental arthritis models. Ideally, the key finding would be confirmed in a separate arthritis model, but considering the extensive amount of data already included in the manuscript, a balanced discussion of that particular limitation of the study appears sufficient.

Reviewer #2 (Breg, B/T interaction)(Remarks to the Author):

This study by Oleinika and colleagues describe a role for CD1d on T2-MZP in suppressing arthritis in mice via their interaction with iNKT cells in an IL-10-independent manner, which is supported by the data shown. Less convincing is the mechanism whereby the suppression occurs. The data shown suggesting that the mechanism is by induction of IFN-g production of iNKT cells and suppression of CD4 T cell differentiation is less convincing. The manuscript is well written and provides an explanation for why B cells that express high levels of CD1d exhibit regulatory function. This result will be of interest to the field.

Specific Comments

1. Figure 3H. The purpose of the arrowheads should be stated in the figure legend.
2. Fig. 5B-C. While the data may be significant the changes observed are so small that they are not likely biologically relevant.
3. Figure 5D. In this figure it is claimed that differentiation of CD4+ T cells is examined. In the results it is stated that the T cells come from mice with arthritis, yet in the figure legend this is not stated. Thus the source of the T cells is not clear. If the T cells were from mice with arthritis then they would have already encountered signals driving their differentiation. The assay as designed does not actually determine whether T cells differentiated. What it shows is whether uMT iNKT cells can promote the production of IFN-g in cD4 T cells of an unknown activation/differentiated status. Thus the data are of little value. In addition, the change in the percentage of IFN-g+ cells is so small that it is not likely biologically relevant.
4. Fig. 5E. This figure does not show that IFN-g inhibited the differentiation of Th1 and Th17 cells. What it shows is that IFN-g inhibited T-bet and ROR-gt expression. This is not the same thing as preventing differentiation. Again the changes are so small that it is not clear if they are biologically relevant.
5. The studies in Fig. 5 do not directly demonstrate that Breg induced IFN-g production by iNKT cells. Thus the claims made in this figure are not supported by the data shown.
6. Figure 8D. The B cells utilized in this assay are not the same as those being examined throughout the rest of the study. Thus the experimental design fails to determine whether iNKT cells are required for the development of T2-MZP that can suppress via interaction with iNKT cells via CD1d. Since the mechanism is IL-10-independent it is not clear why IL-10 producing B cells were examined. It is not clear whether IL-10-producing B cells that are not CD1d+T2-MZP are required for suppression in the studies shown.
7. Line 328. B10 are not a definitive Breg subset as suggested. They are simply IL-10 producing B cells.

Minor comments

8. Some of the journal names in the reference list are not abbreviated.
9. Isolation of liver cells is described in the Materials and Methods, but it is not clear where these cells were used in the figures.

Reviewer #3 (Breg, tolerance)(Remarks to the Author):

This paper aims to characterise a novel, IL-10-independent mechanism by which CD1d+ B cells present lipid to iNKT cells which in turn, through IFN γ release, inhibit Th1 and Th17 immune responses in a mouse model of arthritis.

The authors make excellent use of chimeric mouse models to address their hypotheses. This is a clearly written, detailed study.

There are issues which should be addressed to clarify and confirm some of the authors' claims. Some of the experiments have only been repeated once.

1a. If Breg are important in this arthritis mouse model, one would expect to see an exacerbation of arthritis in the uMT mouse compared to the WT, regardless of a-GalCer. Can the authors explain why this does not appear to be the case?

1b. It would be relevant here to show the iNKT levels within the draining lymph node, as well as the spleen.

To claim that there is an equal increase in iNKT levels of uMT and WT mice, this increase between the two models needs to be compared in the statistical analysis. Currently, only the increase in iNKT when a-GalCer is added within each model, is shown.

1e. A substantial proportion of iNKT cells still express IFN γ in the uMT model – what about other cytokines, such as IL-4 and IL-10 etc?

1g. The authors should add a label to the bar chart indicating no significance between WT and uMT liver IFN γ +iNKT when a-GalCer is added.

2c. To claim that there is an equal increase in iNKT levels of WT and Rituximab-treated mice, this increase between the two conditions needs to be compared in the statistical analysis. Currently, only the increase in iNKT when a-GalCer is added within each model, is shown.

S1a-d. There is a substantial decrease in iNKT cells to negligible levels, when DCs are depleted and a-GalCer is added. Can the authors explain how such a decrease in iNKT levels still does not affect the amelioration of arthritis when a-GalCer is added upon DC-depletion? One would expect some clinical difference, given the complete lack of iNKT cells upon a-GalCer administration in this model, as this would surely translate to negligible levels of iNKT cells expressing IFN γ ?

Fig. S2. CD1d expression within the IL-10 transgenic mice should be demonstrated.

Fig. 4d. Can this increase in gene expression of IL-10, IL-21 etc be detected as proteins i.e. ELISA/Facs, to support functional relevance?

Fig. 5a-b. When anti-IFN γ is added to WT with vehicle (positive control), it appears that inflammation and IFN γ +CD4+ T cell levels are reduced compared to WT with vehicle and no anti-IFN γ (negative control), indicating that the anti-IFN γ itself is having an effect on inflammation. Thus it is not possible to deduce that the increased inflammation upon a-GalCer therapy in the presence of anti-IFN γ is directly due to blockade of IFN γ normally elicited by a-GalCer. i.e. The effect on IFN γ expression by T cells may be due to the fact that IFN γ has been blocked by the inhibitor, rather than the action of a-GalCer being blocked by the inhibitor.

The bar chart in Fig. 5a also does not represent what is described in the legend. In Fig. 5b-c, representative dot plots should be provided, particularly given such low percentages.

Fig. 5c. It is not clear what the statistical bars on the graph are comparing. Why is there such a large relative increase in ROR- γ t+ T cells by uMT mice when a-GalCer is added? What non-B cells express CD1d in this model?

Fig. 5d. To directly show that iNKT cells can inhibit IFN γ expression in CD4+ T cells via an IFN γ -dependent mechanism, the authors should isolate iNKT cells from WT mice treated with anti-IFN γ and culture these IFN γ blocked iNKT cells with T cells from arthritic WT mice.

Fig. 5e. Given that % levels have been demonstrated throughout the manuscript, it is not clear why the authors choose to illustrate MFI. % levels should be included for consistency.

S1f. Given that this model is set up differently to the model described in Fig 1-2, is there an overall difference in levels of IFN γ +iNKT cells between the uMT model and DC-depletion model?

S4c. The x axis on the second facs plot showing pre-sort strategy should read, I assume, 'CD23,' not 'CD24.' Whilst Facs staining is shown, purities are not actually stated, this should be amended.

Fig 6. Whilst this figure is interesting and suggests iNKT-dependency of T2MZP Breg, some clarity is required. Given that T2MZP Breg-mediated suppression has been previously shown to be IL-10-dependent in similar models (by the same group), they should demonstrate the IL-10 expression by T2MZP Breg in both WT and CD1d $^{-/-}$ mice. Is there any difference in IL-10 expression to account for the different clinical outcomes? Also, when T2MZP Breg are transferred into WT mice, is there a decrease in IFN γ expression by iNKT cells but maintenance of frequency, as described earlier in the paper? This needs to be established before extrapolating that T2MZP Breg represent the earlier described CD1d $^{+}$ Breg.

Fig 6b, 6d, 6e. The statistics on the bar charts are poorly labelled and it is unclear why different experimental groups are being compared upon different charts. On some graphs, the statistics are completely absent, despite comments within the text.

Fig. 7b-e. The text states that the levels of iNKT cells Ki-67, PLZF and IFN γ +iNKT cells are comparable between control and MZ B cell-depleted mice. The statistics on the graphs compare a-GalCer treatment to vehicle only, and do not compare the levels between the control and experimental groups. This should be amended.

Given that MZ B cells express significantly higher levels of CD1d than any other splenic B cell subset (as demonstrated in the supplementary data), and the authors claim that suppression is mediated by CD1d expressing B cells, why cannot MZ B cells suppress inflammatory responses similar to T2MZP Breg or CD1d $^{+}$ B cells? The suggestion that MZ B cells may not come in contact with iNKT cells in Fig. 3h, is associative at best.

Fig. 8a-c. Representative Facs plots of IL-10 staining should be shown.

Given that a-GalCer presentation by B cells to iNKT cells is known to result in differentiation of ab-producing B cells, the authors should look to see whether a) this is the case in this model, b) ab-producing B cells/plasma cells are subsequently regulatory and express IL-35.

We thank all the referees for their insightful comments. We believe and hope that we have addressed all the concerns raised and that the manuscript is now acceptable for publication in Nature Communications.

Our responses to the referees' comments are reported in red. We have performed new experiments and they are now included as figures 5D, 6G, S1B, S2E, S3D, S4D.

Reviewers' comments:

Reviewer #1 (NKT, CD1d, Breg)(Remarks to the Author):

This manuscript has investigated interactions between CD1d-expressing regulatory B cells and iNKT cells. Using a rheumatoid arthritis model in mice, the authors show that CD1d-mediated antigen presentation by Bregs is critical for iNKT cell function. These interactions were required for the suppressive effects of iNKT cell activation on arthritis. These effects were independent of IL-10 production by Breg cells. From these findings, the authors conclude a critical role of CD1d on regulatory B cells for regulating iNKT cell function, in a manner that is independent of IL-10 production by B cells.

General comments:

The manuscript includes an impressive amount of data, with a variety of complementary approaches such as cell depletion, knockout mice, mixed bone marrow chimeras, adoptive transfers, etc. The data are convincing, well-described, and potentially important for understanding interactions between regulatory B cells and iNKT cells during autoimmunity and inflammatory responses.

Minor comments:

1. Mizoguchi et al. (Immunity 2002; 16:219-230) were the first to provide evidence for a role of CD1d on B cells in controlling inflammatory responses in a colitis model. In this system, CD1d and IL-10 were important, but not NKT cells. This report should be included and differences with the present work may be briefly discussed. For example, in some cases signaling through CD1d (e.g., via antibody stimulation) has been shown to influence functions of CD1d-expressing cells, providing one potential explanation for CD1d-dependent but NKT cell-independent effects (I believe the Mizoguchi paper largely used B cells purified based on CD1d expression, presumably using mAbs that could have potentially activated the B cells). **We have now cited the work from Mizoguchi *et al.* and discussed differences with our work.**

2. A variety of prior studies have studied the role of iNKT cells and iNKT cell activation on different models of rheumatoid arthritis. This includes Coppieters et al. (J. Immunol. 2007; 179:2300-2309); Horikoshi et al. (PLoSOne 2012; e51215); Miellot et al. (EJI 2005; 35:3704-3713), and several others. Although the model employed here is different than most of the models employed in these prior studies, these studies should be briefly discussed, because the studies reported here might not apply to other experimental arthritis models. Ideally, the key finding would be confirmed in a separate arthritis model, but considering the extensive amount of data already included in the manuscript, a

balanced discussion of that particular limitation of the study appears sufficient. **We have now discussed the other studies and included the potential limitations of our model.**

Reviewer #2 (Breg, B/T interaction)(Remarks to the Author):

This study by Oleinika and colleagues describe a role for CD1d on T2-MZP in suppressing arthritis in mice via their interaction with iNKT cells in an IL-10-independent manner, which is supported by the data shown. Less convincing is the mechanism whereby the suppression occurs. The data shown suggesting that the mechanism is by induction of IFN- γ production of iNKT cells and suppression of CD4 T cell differentiation is less convincing. The manuscript is well written and provides an explanation for why B cells that express high levels of CD1d exhibit regulatory function. This result will be of interest to the field.

Specific Comments

1. Figure 3H. The purpose of the arrowheads should be stated in the figure legend. **We have now corrected this.**

2. Fig. 5B-C. While the data may be significant the changes observed are so small that they are not likely biologically relevant. **Just as an example, the results in Figure 5B show a 70% reduction in IFN- γ -producing cell frequency after α -GalCer treatment compared to only 42% when IFN- γ is neutralised. It is always difficult to extrapolate the biological importance of small yet significant differences observed *in vitro*, however in this case the results are supported by the *in vivo* results shown in Figure 5A.**

3. Figure 5D. In this figure it is claimed that differentiation of CD4⁺T cells is examined. In the results it is stated that the T cells come from mice with arthritis, yet in the figure legend this is not stated. Thus the source of the T cells is not clear. If the T cells were from mice with arthritis then they would have already encountered signals driving their differentiation. The assay as designed does not actually determine whether T cells differentiated. What it shows is whether uMT iNKT cells can promote the production of IFN- γ in CD4 T cells of unknown activation/differentiation. Thus the data are of little value. In addition, the change in the percentage of IFN- γ + cells is so small that it is not likely biologically relevant. **T cells were isolated from arthritic mice. This has now been added in the figure legend. All T cells were isolated 16h following i.a. injection with mBSA, now stated in the materials and methods. The referee is correct in stating that we cannot say that there is a suppression of T cell differentiation. What our data show is that iNKT cells suppress the expression of IFN- γ in CD4⁺ T cells. This has now been corrected. However, we disagree with the last comment that the data are of little value. The results in this figure show that WT and not μ MT iNKT cells activated *in vivo* can significantly inhibit IFN- γ production by T cells directly.**

4. Fig. 5E. This figure does not show that IFN- γ inhibited the differentiation of Th1 and Th17 cells. What it shows is that IFN- γ inhibited T-bet and ROR- γ t expression. This is not the same thing as preventing differentiation. Again the changes are so small that it is not clear if they are biologically relevant. **The referee is correct and we apologise for having used the wrong terminology. The results in this Figure 5E show that IFN- γ**

inhibited T-bet and ROR- γ t expression.

5. The studies in Fig. 5 do not directly demonstrate that Breg induced IFN-g production by iNKT cells. Thus the claims made in this figure are not supported by the data shown. We thank the referee for this comment and have amended the introductory sentence (line 239) to reflect the aim of this figure. This figure was never intended to show that Bregs induce iNKT cells producing IFN- γ , but to address whether IFN- γ produced by iNKT cells could regulate CD4⁺ T cell expression of IFN- γ and IL-17, as well as T-bet and ROR- γ t (master regulators of Th1/Th17 subsets). Of note we have performed a new experiment now reported in Figure 5D following the suggestion of referee 3. In the experiment we isolated iNKT cells from α -GalCer treated μ MT and WT mice at 16h when we observe highest IFN- γ production by iNKT cells. We then cultured these with CD4⁺ T cells from WT mice (these match the results in figure 5D previously). To demonstrate that IFN- γ directly had an effect on CD4⁺ T cell suppression we then blocked IFN- γ *in vitro*, and observed that the suppression of CD4⁺ T cell IFN- γ by WT iNKT cells was abrogated.

Figure 8D. The B cells utilized in this assay are not the same as those being examined throughout the rest of the study. Thus the experimental design fails to determine whether iNKT cells are required for the development of T2-MZP that can suppress via interaction with iNKT cells via CD1d. Following the suggestion of this referee we have removed panel 8D as we have already shown in Figure 6C and D that T2-MZP B cells from *Ja18*^{-/-} mice suppress disease in WT mice to the same extent as WT T2-MZP, demonstrating that in the absence of iNKT cells T2-MZP Breg development is normal.

Since the mechanism is IL-10-independent it is not clear why IL-10 producing B cells were examined. To address whether Bregs (IL-10 is a marker of Bregs) develop normally in the absence of iNKT cells. If this referee thinks that the last figure is confusing we will be happy to remove it.

7. Line 328. B10 are not a definitive Breg subset as suggested. They are simply IL-10 producing B cells. We have specified that B10 are IL-10 producing B cells.

Minor comments

8. Some of the journal names in the reference list are not abbreviated. We have now corrected this.

9. Isolation of liver cells is described in the Materials and Methods, but it is not clear where these cells were used in the figures. Liver cells were used in Figure 1, F and G.

Reviewer #3 (Breg, tolerance)(Remarks to the Author):

This paper aims to characterise a novel, IL-10-independent mechanism by which CD1d⁺ B cells present lipid to iNKT cells which in turn, through IFN γ release, inhibit Th1 and Th17 immune responses in a mouse model of arthritis. The authors make excellent use of

chimeric mouse models to address their hypotheses. This is a clearly written, detailed study. There are issues which should be addressed to clarify and confirm some of the authors' claims. Some of the experiments have only been repeated once. **All our experiments have been performed a minimum of two times. The only experiment that has been performed once is the RNA-seq, however the genes found to be significantly upregulated and of immunological interest were validated by PCR.**

1a. If Breg are important in this arthritis mouse model, one would expect to see an exacerbation of arthritis in the uMT mouse compared to the WT, regardless of α -GalCer. Can the authors explain why this does not appear to be the case? **B cells play pleiotropic roles in the immune system including a protective role mediated by Bregs, but also a pathogenic role mediated by antigen presentation to T cells, production of antibody and secretion of pro-inflammatory cytokines. The results in figure 7 showing that depletion of MZ B cells leads to a milder arthritis compared to non-depleted suggest that MZ B cells play a pathogenic role in this disease. Thus, we propose that in the absence of pathogenic and protective B cells we do not observe an exacerbation of disease.**

1b. It would be relevant here to show the iNKT levels within the draining lymph node, as well as the spleen. To claim that there is an equal increase in iNKT levels of uMT and WT mice, this increase between the two models needs to be compared in the statistical analysis. Currently, only the increase in iNKT when α -GalCer is added within each model, is shown. **We have now included representative flow cytometry plots and bar charts (Figure S1A) showing the frequency and absolute number of iNKT cells in the draining lymph node. We apologize for not being sufficiently clear on how the statistical analysis was performed. We have now added the statistics for the comparisons of the α -GalCer treated groups.**

1e. A substantial proportion of iNKT cells still express IFN γ in the uMT model – what about other cytokines, such as IL-4 and IL-10 etc? **We have now included new results showing other cytokines produced by splenic iNKT cells isolated from μ MT and WT mice treated with α -GalCer or vehicle. The new panel included in Figure S1B shows that in response to α -GalCer iNKT cells from μ MT mice produce less IL-4, compared to WT iNKT cells, whereas no differences were detected in IL-10, IL-13, or GM-CSF between α -GalCer-treated μ MT and WT iNKT cells.**

1g. The authors should add a label to the bar chart indicating no significance between WT and uMT liver IFN γ +iNKT when α -GalCer is added. **We have added a label indicating that these are not significantly different.**

2c. To claim that there is an equal increase in iNKT levels of WT and Rituximab-treated mice, this increase between the two conditions needs to be compared in the statistical analysis. Currently, only the increase in iNKT when α -GalCer is added within each model, is shown. **We have added a label indicating that these are not significantly different.**

S1a-d. There is a substantial decrease in iNKT cells to negligible levels, when DCs are

depleted and α -GalCer is added. Can the authors explain how such a decrease in iNKT levels still does not affect the amelioration of arthritis when α -GalCer is added upon DC-depletion? One would expect some clinical difference, given the complete lack of iNKT cells upon α -GalCer administration in this model, as this would surely translate to negligible levels of iNKT cells expressing IFN γ ? We apologise as we have incorrectly stated the time-points in this particular figure. The proliferation of iNKT cells is measured at 72h, whereas the production of IFN- γ at 16h following α -GalCer administration. We have now included the results here for the referee perusal showing that frequencies of iNKT cells at 16h are comparable in control and CD11c⁺ cell depleted mice treated with α -GalCer.

Fig. S2. CD1d expression within the IL-10 transgenic mice should be demonstrated. We believe that the referee is referring to the B cell IL-10-deficient chimeric mice. We have included in Figure S2E results showing that there is no difference between the expression of CD1d on B cells or on any other cells between the chimeric and the WT mice. Similar results were shown in the IL-10 deficient mice (data not shown).

Fig. 4d. Can this increase in gene expression of IL-10, IL-21 etc be detected as proteins i.e. ELISA/FACS, to support functional relevance? We have included new results in Figure S3D showing increased frequencies of IL-10⁺ and IL-13⁺ iNKT cells in the absence of CD1d-presenting B cells in α -GalCer treated mice, whereas no positive staining in iNKT cells was found for IL-9, IL-17, IL-21 (data not shown). A trend in the reduction in the frequency of TNF- α ⁺ iNKT cells in response to α -GalCer treatment was also observed.

Fig. 5a-b. When anti-IFN γ is added to WT with vehicle (positive control), it appears that inflammation and IFN γ +CD4⁺ T cell levels are reduced compared to WT with vehicle and no anti-IFN γ (negative control), indicating that the anti-IFN γ itself is having an effect on inflammation. Thus it is not possible to deduce that the increased inflammation upon α -GalCer therapy in the presence of anti-IFN γ is directly due to blockade of IFN γ normally elicited by α -GalCer. i.e. The effect on IFN γ expression by T cells may be due to the fact that IFN γ has been blocked by the inhibitor, rather than the action of α -GalCer being blocked by the inhibitor.

We disagree with this interpretation of our results, if α -GalCer is not working through anti-IFN- γ , treatment with both should lead to greater suppression than either alone. The results show that blocking IFN- γ reduces the suppressive capacity of α -GalCer compared to the group of mice that have been treated with isotype control+ α -GalCer. Therefore, a significantly greater suppression is achieved by α -GalCer treatment alone than in mice treated with anti-IFN- γ + α -GalCer. These data support data in figure 5A, where we show that blocking IFN- γ reduces the suppressive capacity of α -GalCer.

We have added the below additional figure (for the referee's perusal) showing that there is up to 76% reduction in the number of CD4⁺ T cells producing IFN- γ in mice treated with isotype control+ α -GalCer. This is abrogated when anti-IFN- γ was administered to α -GalCer treated mice. Similarly for the number of IL-17⁺ CD4⁺ T cells, a 54% reduction is observed in isotype-control treated mice, while complete loss of suppression of α -GalCer in anti-IFN- γ treated mice.

The bar chart in Fig. 5a also does not represent what is described in the legend. We would be grateful if the referee could advise what is specifically incorrect in this figure legend as we have checked and it appears correct.

In Fig. 5b-c, representative dot plots should be provided, particularly given such low percentages. We have now added representative plots.

Fig. 5c. It is not clear what the statistical bars on the graph are comparing. We apologize if we have not been sufficiently clear. We have now added the statistical analysis reporting also those comparisons that were not significantly different.

Why is there such a large relative increase in ROR-gt⁺ T cells by μ MT mice when α -GalCer is added? What non-B cells express CD1d in this model? This could be explained by the lack of B cell-induced iNKT cells with suppressive capacity. Other CD1d expressing cells are driving the inflammatory response. Figure S4A shows that other cells beside B cells express CD1d.

Fig. 5d. To directly show that iNKT cells can inhibit IFN γ expression in CD4⁺ T cells via an IFN γ -dependent mechanism, the authors should isolate iNKT cells from WT mice treated with anti-IFN γ and culture these IFN γ blocked iNKT cells with T cells from arthritic WT mice. Following the advice of this referee we have performed the following experiment: we isolated iNKT cells from α -GalCer treated μ MT and WT mice at 16h

when we observe highest IFN- γ production by iNKT cells. We then cultured these with CD4⁺ T cells from WT mice (these match the results in figure 5D previously). To demonstrate that IFN- γ directly had an effect on CD4⁺ T cell suppression we then blocked IFN- γ *in vitro*, and observed that the suppression of CD4⁺ T cell IFN- γ by WT iNKT cells was abrogated. The reason why we block IFN- γ *in vitro* rather than *in vivo* (as suggested) was to ensure that the IFN- γ produced by iNKT cells is neutralized. Blocking IFN- γ *in vivo* would not stop iNKT cells to release it *in vitro*. These results have additional conditions to the original Figure 5D and thus we have now replaced these results with the present one.

Fig. 5e. Given that % levels have been demonstrated throughout the manuscript, it is not clear why the authors choose to illustrate MFI. % levels should be included for consistency. **We have now substituted the MFI with the % levels in Figure 5E.**

S1f. Given that this model is set up differently to the model described in Fig 1-2, is there an overall difference in levels of IFN γ ⁺iNKT cells between the uMT model and DC-depletion model? **We are not entirely sure what this referee is referring to. As shown in Figure 2E versus Figure S1H, our data show that the frequency of IFN-g producing iNKT cells is reduced in the absence of B cells and not in the absence of DCs.**

S4c. The x axis on the second FACS plot showing pre-sort strategy should read, I assume, 'CD23,' not 'CD24.' Whilst FACS staining is shown, purities are not actually stated, this should be amended. **We have now corrected both.**

Fig 6. Whilst this figure is interesting and suggests iNKT-dependency of T2MZP Breg, some clarity is required. Given that T2MZP Breg-mediated suppression has been previously shown to be IL-10-dependent in similar models (by the same group), they should demonstrate the IL-10 expression by T2MZP Breg in both WT and CD1d^{-/-} mice. Is there any difference in IL-10 expression to account for the different clinical outcomes? **We have now included representative staining and summary data showing the frequency of IL-10⁺ T2-MZP B cells in *Cd1d*^{-/-} and WT mice, Figure S4D. The results show that there is no difference in IL-10 production by T2-MZP B cells from these mice.**

Also, when T2MZP Breg are transferred into WT mice, is there a decrease in IFN γ expression by iNKT cells but maintenance of frequency, as described earlier in the paper? This needs to be established before extrapolating that T2MZP Breg represents the earlier described CD1d⁺ Breg. **We have included new data (Figure 6G) showing that the transfer of WT T2-MZP, but not *Cd1d*^{-/-} T2-MZP B cells leads to an increase in iNKT cells producing IFN- γ .**

Fig 6b, 6d, 6e. The statistics on the bar charts are poorly labelled and it is unclear why different experimental groups are being compared upon different charts. On some graphs, the statistics are completely absent, despite comments within the text. **We have now amended the statistics.**

Fig. 7b-e. The text states that the levels of iNKT cells Ki-67, PLZF and IFN γ ⁺iNKT cells

are comparable between control and MZ B cell-depleted mice. The statistics on the graphs compare a-GalCer treatment to vehicle only, and do not compare the levels between the control and experimental groups. This should be amended. **We have now corrected this.**

Given that MZ B cells express significantly higher levels of CD1d than any other splenic B cell subset (as demonstrated in the supplementary data), and the authors claim that suppression is mediated by CD1d expressing B cells, why cannot MZ B cells suppress inflammatory responses similar to T2MZP Breg or CD1d⁺ B cells? The suggestion that MZ B cells may not come in contact with iNKT cells in Fig. 3h, is associative at best. **As for T cells, iNKT cell threshold of activation and functional polarization may be dictated by the signal strength provided by the APC. It is possible that MZ due to the high expression of CD1d provide a stronger signal than T2-MZP, which influences the net outcome of iNKT cell response. I hope that this referee agrees with us that unravelling the mechanisms by which MZ modulates iNKT cells responses is beyond the scope of this manuscript. Nevertheless, as this referee thinks that our explanation is associative, we have removed this sentence from the results section.**

Fig. 8a-c. Representative FACS plots of IL-10 staining should be shown. **We have now added representative flow cytometry plots for Figure 8, A and B. C is data from an ELISA analysis further confirming the amount of IL-10 produced.**

Given that a-GalCer presentation by B cells to iNKT cells is known to result in differentiation of ab-producing B cells, the authors should look to see whether a) this is the case in this model, b) ab-producing B cells/plasma cells are subsequently regulatory and express IL-35. **Following the suggestion of this referee we have performed the experiments and we report the following: a) we do not observe induction of CD138⁺ plasmablasts following α -GalCer administration; b) we can not detect any changes in Ebi3 or p35 mRNA by qPCR in B cells (data not shown).**

These results were not entirely surprising as we have previously published that mice lacking p35 develop milder AIA compared to WT mice (1). The decrease in disease severity was mirrored by an increase in T2-MZP Bregs, and a decrease in MZ B cells (further supporting the pathogenic effect of this population of B cells in arthritis). In the same paper, we also show that the absence of p35 leads to an increased availability of Ebi3, consequential binding to p28 and release of IL-27, which mediates this suppression. We have also shown that T2-MZP, or B cells stimulated with anti-CD40, isolated from p35-deficient mice protect upon adoptive transfer to recipient mice. Therefore, in this system IL-35 does not seem to play a regulatory function. We have added a sentence mentioning these results in the discussion.

1. Vasconcellos R, Carter NA, Rosser EC, Mauri C. IL-12p35 subunit contributes to autoimmunity by limiting IL-27-driven regulatory responses. *Journal of immunology*. 2011;187(6):3402-12.

Reviewers' comments:

Reviewer #1 (Remarks to the Author):

The authors have responded appropriately to all reviewer comments, resulting in a much improved manuscript.

Reviewer #2 (Remarks to the Author):

The authors have sufficiently revised the application.

Reviewer #3 (Remarks to the Author):

The authors have added some new data to improve the manuscript. However, there are still questions from the initial review which have not yet been answered, as well as some concerning issues which need to be addressed.

General comments:

All FACS plots for CD1d, IL-10, IFN γ , IL-17 etc should have accompanying isotype or FMO controls stains displayed in the figures to demonstrate how gating was performed.

All data provided by the authors in their response should be incorporated in the manuscript. This is not currently the case.

Some of the main experiments, in particular, in Fig. 5 & 6, have only been performed twice – given such marginal differences in % and frequency of IFN γ + and IL-17+ T cells between groups in these figures, it is difficult to see how statistical significance has been reached – whilst numbers per experimental groups are between 3-4 in some experiments (i.e. Fig. 6b, d, f), these are technical controls within the group and are not true repeats; thus they cannot be included as individual experimental results within an analysis to ascertain statistical significance across repeated experiments. If an experiment has been performed twice, then ordinarily the overall result from each of these experiments is compared to the each other. Thus an explanation of how statistical significance could have been reached when looking at the bar charts of cytokine expression within these figures. In addition, it is probably unlikely that such marginal changes in very low levels of cytokine expression could have any functional or biological significance. Thus the data as presented do not appear to support the claims made.

Specific comments:

1a. The authors argue that Fig 7 demonstrates that MZ depletion results in a milder form of arthritis, and thus these apparently pleiotropic effects of B cells can be extrapolated to explain why uMT mice do not present with an exacerbated form of arthritis when compared to WT. However, Fig. 7a does not demonstrate any clinical difference in disease severity when comparing WT to MZ-depleted mice in the presence of a vehicle. Thus the original question from the first review remains unanswered.

1e. IFN γ expressed by iNKT cells decreases by only about 1/3 in the uMT model when compared to the WT – Over 40% of iNKT still express IFN γ in the uMT mouse. Yet the authors argue that it is this

decrease that is principally responsible for the clinical change displayed in Fig 1a.

Can the authors account for this? If arthritis is principally suppressed by IFN γ expressed by iNKT (argued throughout the manuscript), which in turn is driven by CD1d $^{+}$ B cells, then how are 40% of iNKT cells still expressing IFN γ and why is this substantial proportion of apparently regulatory cells not able to continue to suppress arthritis in the uMT model? Is there a dose effect?

S1a-d. Why is there no exacerbation of disease at 72 hrs following DC depletion, when numbers of iNKT cells have been demonstrated to be negligible at this time-point? Furthermore, why do the authors demonstrate iNKT cell frequency (at 72 hrs) and IFN γ expression (at 16 hrs) at different time-points in both Fig.1 and S1, particularly since at 72 hrs, there is little clinical difference in arthritis between experimental groups – i.e. at this time-point, all mice appear to be in remission. How is the 72 hr time-point relevant in this context?

Fig. 3a-b. The rationale for measuring T cell cytokine expression measured at d7, when disease state is not demonstrated at this time-point should be explained. According to the data shown, clinical severity scores are only demonstrated to d3. Also, given that iNKT cell expression of IFN γ has been consistently demonstrated at 16hrs (here and indeed in the previous figures) at about the time-point when the greatest difference in disease score is seen between clinical groups, surely the more relevant measurement of T cell cytokine expression at this earlier time-point should be demonstrated. Throughout the manuscript, IFN γ expression by iNKT cells is measured at 16 hrs, iNKT cell frequencies at d3 and T cell cytokine expression at d7. It is difficult to see how based on these disparate time-points, two of which do not correlate with clinical disease progression, the authors form their hypotheses of cause and effect. Further clarification required.

S3d. The authors describe a 'trend' in reduction of frequency of TNF α^{+} iNKT cells in response to aGalCer – given that no statistical significance was found, such statements are misleading and as presented appear to have no scientific basis.

Fig. 5. In response to the initial review, the authors argue that if aGalcer is not working through anti-IFN γ , treatment with both should lead to greater suppression than either alone. This assumption is not supported by the data currently provided by the authors. Whilst Fig 5d does display the results of a well-designed experiment, the claims made by the authors are simply not supported by the data and the other experiments provided in this figure – particularly given the extremely small changes in very low levels of IFN γ and IL-17 expression by CD4 $^{+}$ T cells – It is difficult to see how such small changes could have any real biological significance. Furthermore, FACS plot isotype or FMO controls should be demonstrated.

Bar chart in Fig. 5a. The legend states that that the chart shows suppression of knee swelling in 4 groups: anti IFN γ -treated and isotype-treated WT receiving a-Galcer, compared to anti IFN γ -treated and isotype-treated WT receiving vehicle alone. But only two groups appear to be compared on the chart, at different time-points.

Fig. 5c-d. A key control is missing: CD4 T cell cytokine expression when splenocytes are cultured with anti-CD3 and anti-IFN γ only.

Fig. 6a. The progression of disease severity of arthritis in CD1d $^{-/-}$ mice should be demonstrated, as additional supplementary data, to demonstrate that this model can be used as a comparator to WT in this context.

Fig. 6b, d, f. FACS dot plot staining should be shown.

In answer to the question asking why MZ B cells cannot suppress inflammatory responses similar to TZMP Breg or CD1d⁺ B cells given that the CD1d MFI of MZ B cells is x3 greater than that of TZMP, the authors suggest that this may be due to activation thresholds of iNKT cells. While this may be the case, the authors have not provided evidence to explain this discrepancy in their findings (particularly given that there is no real clinical difference in arthritis severity upon MZ depletion in this model (Fig 7a)). The title and the entire premise of the manuscript hinges on the claim that iNKT cells can suppress arthritis via IFN γ production, and that this is in turn driven by the CD1d presented by B cells. Thus the authors must provide an explanation with supporting data, for why there is quite a significant exception to this rule in the form of the MZ B cell population, in order to convincingly validate the mechanisms that they claim.

Royal Free and University College Medical School
University College London
Department of Medicine
Centre For Rheumatology Research
5 University Street, London
WC1E 6JF, UK

Claudia Mauri (PhD)
Professor of Immunology
Vice-Dean International Faculty of Medical Science
Telephone 44 (0) 2031082155
E-mail: c.mauri@ucl.ac.uk

London, 2nd November 2017

We thank referee 1 and 2 for having found our manuscript suitable for publication.

Reply to referee 3 comments:

We thank the referee for the further comments. Please find below our reply point by point to the comments raised. We hope that this referee agrees with us that our manuscript contains several very novel findings (above all that Bregs can suppress in an IL-10 independent manner and that CD1d play a pivotal role in this suppression). Like in the majority of novel studies our findings have raised new questions that will be address by us and other groups in future investigations.

General comments

All Facs plots for CD1d, IL-10, IFN γ , IL-17 etc should have accompanying isotype or FMO controls stains displayed in the figures to demonstrate how gating was performed.

These have now been included throughout the figures.

All data provided by the authors in their response should be incorporated in the manuscript. This is not currently the case.

These have now been included in Figures S1I and S3E.

Some of the main experiments, in particular, in Fig. 5 & 6, have only been performed twice – given such marginal differences in % and frequency of IFN γ + and IL-17+ T cells between groups in these figures, it is difficult to see how statistical significance has been reached – whilst numbers per experimental groups are between 3-4 in some experiments (i.e. Fig. 6b, d, f), these are technical controls within the group and are not true repeats; thus they cannot be included as individual experimental results within an analysis to ascertain statistical significance across repeated experiments. If an experiment has been performed twice, then ordinarily the overall result from each of these experiments is compared to the each other. Thus an explanation of how statistical significance could have been reached when looking at the bar charts of cytokine expression within these figures. In addition, it is probably unlikely that such marginal changes in very low levels of cytokine expression could have any functional or biological significance. Thus the data as presented do not appear to support the claims made.

We have consulted with our statistician at UCL, who has confirmed that our statistical analysis is correct. For example, in Figure 6 we are not testing whether T2-MZP B cell transfer can inhibit arthritis, which may require multiple repeats with separate pooled T2-MZP B cells to check that donor cells are consistently comparable and suppressive between experiments. Indeed, the suppressive effect of T2-MZP B cells has been published multiple times.

Here we are testing the effect on individual recipient mice of transfer of known suppressive cells, each individual recipient mouse is a biologically discrete entity and therefore a true biological repeat, not an experimental one. A technical control in this context would be to, for instance, split

the T cells of one recipient mouse into three and perform the same flow cytometry staining on the three T cell samples to get three separate readouts for, say, IFN- γ . This is not how this experiment was performed and therefore we believe that our analysis is correct.

Specific comments:

1a. The authors argue that Fig 7 demonstrates that MZ depletion results in a milder form of arthritis, and thus these apparently pleiotropic effects of B cells can be extrapolated to explain why uMT mice do not present with an exacerbated form of arthritis when compared to WT. However, Fig. 7a does not demonstrate any clinical difference in disease severity when comparing WT to MZ-depleted mice in the presence of a vehicle. Thus the original question from the first review remains unanswered.

We have now included statistical analysis here showing that MZ B cell-depleted mice develop a moderately milder disease compared to WT control mice (still significantly different). To be noted, we only depleted temporally MZ B cells and it is possible that a prolonged treatment could have a “stronger” effect on disease course. Therefore, our original explanation, in particular in the context of μ MT mice, which lack both “bad” and “good” B cells, as well as having other immunological defects as previously reported, still stands.

1e. IFN γ expressed by iNKT cells decreases by only about 1/3 in the uMT model when compared to the WT – Over 40% of iNKT still express IFN γ in the uMT mouse. Yet the authors argue that it is this decrease that is principally responsible for the clinical change displayed in Fig 1a.

Can the authors account for this? If arthritis is principally suppressed by IFN γ expressed by iNKT (argued throughout the manuscript), which in turn is driven by CD1d⁺ B cells, then how are 40% of iNKT cells still expressing IFN γ and why is this substantial proportion of apparently regulatory cells not able to continue to suppress arthritis in the uMT model? Is there a dose effect?

It is important to emphasize that we have never stated that iNKT cells, driven by CD1d⁺ B cells, work exclusively via IFN- γ . We say that this effect is partial. Indeed the results in figure 4 show that nearly 4000 genes are significantly affected when iNKT cells differentiate in the absence of CD1d+B cells. To make sure that our message is consistent with our results we have modified the abstract to explicitly say that CD1d lipid presentation by Bregs is critical for the induction of iNKT cells that, partially, **but not exclusively**, via the release of IFN- γ down-regulate T helper (Th) 1 and Th17 adaptive immune responses and ameliorate arthritis.

As suggested by this referee it is likely that the lack of suppression is due to the reduced number of iNKT cells producing IFN- γ . Unfortunately, it is technically impossible to address this question. The results in Figure 5E show that IFN- γ can suppress CD4⁺ T cell cytokine production in a dose-dependent manner, lending further support to this hypothesis. For clarification we have added a

sentence in the discussion, in which we specify that additional mechanisms may operate, and we have provided a possible explanation (dose response effect) of why the remaining iNKT cells fail to suppress.

S1a-d. Why is there no exacerbation of disease at 72 hrs following DC depletion, when numbers of iNKT cells have been demonstrated to be negligible at this time-point? Furthermore, why do the authors demonstrate iNKT cell frequency (at 72 hrs) and IFN γ expression (at 16 hrs) at different time-points in both Fig.1 and S1, particularly since at 72 hrs, there is little clinical difference in arthritis between experimental groups – i.e. at this time-point, all mice appear to be in remission. How is the 72 hr time-point relevant in this context?

Fig. 3a-b. The rationale for measuring T cell cytokine expression measured at d7, when disease state is not demonstrated at this time-point should be explained. According to the data shown, clinical severity scores are only demonstrated to d3. Also, given that iNKT cell expression of IFN γ has been consistently demonstrated at 16hrs (here and indeed in the previous figures) at about the time-point when the greatest difference in disease score is seen between clinical groups, surely the more relevant measurement of T cell cytokine expression at this earlier time-point should be demonstrated. Throughout the manuscript, IFN γ expression by iNKT cells is measured at 16 hrs, iNKT cell frequencies at d3 and T cell cytokine expression at d7. It is difficult to see how based on these disparate time-points, two of which do not correlate with clinical disease progression, the authors form their hypotheses of cause and effect. Further clarification required.

We apologies for the confusion which we hope now have clarified. We have added a new experiment in Figure S8A and B showing that at 16h iNKT cells produce IFN- γ , but express very little Ki-67; at 72h this is reversed, iNKT cells express high levels of Ki-67, but very little IFN- γ . This should clarify why we measure iNKT cell IFN- γ and proliferation at 16 and 72h, respectively. In addition, for further clarification, we have also included new data showing the development of arthritis over 7 days (Figure S8C), as well as the frequency of IFN- γ ⁺ and IL-17⁺ CD4⁺ T cells at days 3 and 7. The results show that the frequency of these subsets of T cells is similar between the two days. This is because at day 7, although mice have less inflammation than at days 1 and 3, they are not in remission, explaining why upon *in vitro* re-stimulation they express equivalent levels of cytokines (Figure S8D).

S3d. The authors describe a ‘trend’ in reduction of frequency of TNF α + iNKT cells in response to aGalCer – given that no statistical significance was found, such statements are misleading and as presented appear to have no scientific basis.

This has now been removed.

Fig. 5. In response to the initial review, the authors argue that if aGalcer is not working through anti-IFN γ , treatment with both should lead to greater suppression than either alone. This assumption is not supported by the data currently provided by the authors. Whilst Fig 5d does display the results of a well-designed experiment, the claims made by the authors are simply not supported by the data and the other experiments provided in this figure – particularly given the extremely small changes in very low levels of IFN γ and IL-17 expression by CD4⁺ T cells – It is difficult to see how such small changes could have any real biological significance.

We have performed the experiment that was suggested in the previous review by this referee. We do not have any further comments. The other two referees were convinced that all results in Figure 5 support our claims. We respectfully disagree that small changes are not biologically relevant. There are plenty of evidence in the literature that indeed small biological changes can dramatically affect immunological response. In this context, we would like to invite the referee to read our paper in nature medicine (rosser et al 2014). In this paper, for example, we show that small variations in the microbiota composition determine the magnitude of the severity of disease. We have adjusted the text to make sure the message – that IFN- γ is not the only mechanism by which B cell driven

iNKT cells suppress – is clear.

Furthermore, FACS plot isotype or FMO controls should be demonstrated.

We have now added representative FMO controls throughout the figures.

Bar chart in Fig. 5a. The legend states that that the chart shows suppression of knee swelling in 4 groups: anti IFN γ -treated and isotype-treated WT receiving α -GalCer, compared to anti IFN γ -treated and isotype-treated WT receiving vehicle alone. But only two groups appear to be compared on the chart, at different time-points.

In this graph we are showing the percentage of suppression by α -GalCer in anti-IFN γ treated mice and isotype-control-treated mice against their respective vehicle-treated control mice, therefore analysis includes four groups, and the panel demonstrates the two comparisons.

Fig. 5c-d. A key control is missing: CD4 T cell cytokine expression when splenocytes are cultured with anti-CD3 and anti-IFN γ only.

We have now added this control in Figure 5D and show that the frequency of IFN γ ⁺CD4⁺ T cells when CD4⁺ T cells are cultured with anti-CD3 and anti-IFN γ is comparable to the frequency in T cells when cultured with anti-CD3 alone. Unfortunately, we do not have this control for panel 5C. This was due to the high number of conditions and the inherent limitation in cells (160 samples were run of the flow cytometer for this particular experiment). We hope that the referee will agree with us that in view of the lack of any non-specific effect reported on fig 5D it is unlikely that the antibody may affect T cells response in C since they were run at the same time and that we do not have to repeat a complex and expensive experiment.

Fig. 6a. The progression of disease severity of arthritis in CD1d^{-/-} mice should be demonstrated, as additional supplementary data, to demonstrate that this model can be used as a comparator to WT in this context.

This has now been added as Figure S4E.

Fig. 6b, d, f. FACS dot plot staining should be shown.

These have now been included.

In answer to the question asking why MZ B cells cannot suppress inflammatory responses similar to TZMP Breg or CD1d⁺ B cells given that the CD1d MFI of MZ B cells is x3 greater than that of TZMP, the authors suggest that this may be due to activation thresholds of iNKT cells. While this may be the case, the authors have not provided evidence to explain this discrepancy in their findings (particularly given that there is no real clinical difference in arthritis severity upon MZ depletion in this model (Fig 7a)). The title and the entire premise of the manuscript hinges on the claim that iNKT cells can suppress arthritis via IFN γ production, and that this is in turn driven by the CD1d presented by B cells. Thus the authors must provide an explanation with supporting data, for why there is quite a significant exception to this rule in the form of the MZ B cell population, in order to convincingly validate the mechanisms that they claim.

Unfortunately, to date there are no means to generate a mouse lacking exclusively T2-MZP B cells, which would be the only way to assess the “weight” that MZ B cells have on the differentiation of iNKT cells with suppressive function.

However, collectively our data show that:

- 1) in the absence of MZ B cells α -GalCer still suppresses and we still observe the differentiation of IFN γ ⁺PLZF⁺ iNKT cells;
- 2) MZ B cells do not localise with iNKT cells, unlike T2-MZP B cells, which were found in close proximity;
- 3) although as a population MZ B cells express higher levels of CD1d (MFI), as shown in S4A, a large proportion of T2-MZP B cells can also express almost equivalent levels of CD1d, as shown in

Figure S4B. Altogether we strongly believe that our data (in particular the MZ B cell-depletion experiments) support our claims.

We hope that the referee will now find our manuscript acceptable for publication.

Best wishes,
Claudia

REVIEWERS' COMMENTS:

Reviewer #3 (Remarks to the Author):

The finding that MZ B cells express the highest levels of CD1d (x3 higher) is still not addressed and explained with data by the authors.

Reviewer #3 (Remarks to the Author):

The finding that MZ B cells express the highest levels of CD1d (x3 higher) is still not addressed and explained with data by the authors.

We are sorry that this referee still feels that we have not addressed his/her comment. However, we believe that our data showing that the depletion of MZ B cells does not affect the suppressive action of α -GalCer, support the statement that T2-MZP B cells, although they express less CD1d, compared to MZ, are important in the differentiation of immunosuppressive iNKT cells. We have nevertheless toned down our conclusions with respect to the role of MZ B cells, since presently it is impossible to delete T2-MZP B cells alone.